# Frataxin gene editing rescues Friedreich's ataxia pathology in dorsal root ganglia organoid-derived sensory neurons

Pietro Giuseppe Mazzara [1,8,9], Sharon Muggeo [1,9], Mirko Luoni[1], Luca Massimino[1], Mattia Zaghi[1], Parisa Tajalli-Tehrani Valverde [1], Simone Brusco[1], Matteo Jacopo Marzi[2], Cecilia Palma[3], Gaia Colasante[1], Angelo Iannielli[1,4], Marianna Paulis [5], Chiara Cordiglieri [6], Serena Gea Giannelli[1], Paola Podini[1], Cinzia Gellera[7], Franco Taroni [7], Francesco Nicassio [2], Marco Rasponi [3] & Vania Broccoli [1,4✉]

Friedreich's ataxia (FRDA) is an autosomal-recessive neurodegenerative and cardiac disorder which occurs when transcription of the FXN gene is silenced due to an excessive expansion of GAA repeats into its first intron. Herein, we generate dorsal root ganglia organoids (DRG organoids) by in vitro differentiation of human iPSCs. Bulk and single-cell RNA sequencing show that DRG organoids present a transcriptional signature similar to native DRGs and display the main peripheral sensory neuronal and glial cell subtypes. Furthermore, when co-cultured with human intrafusal muscle fibers, DRG organoid sensory neurons contact their peripheral targets and reconstitute the muscle spindle proprioceptive receptors. FRDA DRG organoids model some molecular and cellular deficits of the disease that are rescued when the entire FXN intron 1 is removed, and not with the excision of the expanded GAA tract. These results strongly suggest that removal of the repressed chromatin flanking the GAA tract might contribute to rescue FXN total expression and fully revert the pathological hallmarks of FRDA DRG neurons.

[1] Division of Neuroscience, San Raffaele Scientific Institute, 20132 Milan, Italy. [2] Center for Genomic Science of IIT@SEMM, Istituto Italiano di Tecnologia (IIT), 20139 Milan, Italy. [3] Department of Electronics, Information & Bioengineering, Politecnico di Milano, 20133 Milan, Italy. [4] National Research Council (CNR), Institute of Neuroscience, 20129 Milan, Italy. [5] Humanitas Clinical and Research Center, 20089 Rozzano, Milano, Italy. [6] National Institute of Molecular Genetics "Romeo e Enrica Invernizzi" - INGM, 20122 Milan, Italy. [7] Unit of Medical Genetics and Neurogenetics, Fondazione IRCCS Istituto Neurologico Carlo Besta, 20133 Milan, Italy. [8] Present address: Department of Neuroscience, The Scripps Research Institute, 92037 La Jolla, CA, USA. [9] These authors contributed equally: Pietro Giuseppe Mazzara, Sharon Muggeo. ✉email: broccoli.vania@hsr.it

Friedreich ataxia (FRDA) is an autosomal-recessive neuro-degenerative and cardiac disorder with an estimated incidence of 1:29,000–50,000[1,2]. While the features of FRDA are progressive limb and gait ataxia, dysarthria, dysphagia, loss of deep tendon reflexes, oculomotor dysfunction, and cardiomyopathy, in some cases patients may present a wider range of symptoms including diabetes mellitus, aggressive sclerosis, visual loss and most commonly, defective hearing[3]. Loss of postural balance and ataxia are caused by the degeneration of dorsal root ganglia (DRG), peripheral nerves and the dentate nucleus in the cerebellum[4,5]. FRDA occurs when transcription of the FXN gene is silenced or severely reduced due to the expansion of the GAA trinucleotide repeats in the first intron of the same gene. The majority of patients are homozygous for an expansion of the GAA tract that can span from 44 to 1700 repeats. By comparison, healthy people typically have between 5 and 30 GAA repeats[6]. The length of the GAA expansion correlates proportionally with the severity of the disease and inversely with the age of onset which can occur from infancy to after 60 years of age[7]. The FXN gene silencing is speculated to result from the formation of non-B DNA structures such, as triplexes, or persistent DNA–RNA hybrid structures which might impede transcription initiation and elongation[8]. However, studies performed in FRDA animal and cellular models, then confirmed in human heart, cerebellum and brain explants, indicated repressive histone modifications in the intronic regions flanking the GAA expansion as an additional cause of gene silencing[9–11]. Particularly, increased di- and tri-methylation of H3K9 and deacetylation of histones H3 and H4 at lysine residues are present around the expanded repeat tract suggesting that heterochromatin-mediated transcriptional repression is one the main cause of FXN silencing.

The FXN gene encodes for the mitochondrial protein frataxin which plays several roles in iron metabolism and respiration[12,13]. In fact, insufficient frataxin protein levels in FRDA results in far-reaching mitochondrial dysfunctions that affect functionality and integrity of the mitochondria and more broadly deregulation of the cell antioxidant defenses[14]. In particular, FXN controls the biogenesis of iron–sulfur (Fe–S) clusters which are essential for the proper function of complexes I, II, and III of the electron transport chain, the citric acid cycle enzyme, aconitase, and many others[15]. Interestingly, reduction of FXN in FRDA is most dramatic in the peripheral nerve roots and DRGs as compared with a milder reduction in the central neural structures[16]. Similarly, FRDA patients present a significantly decreased activity of iron–sulfur (Fe–S) proteins in DRGs and, to a lesser extent, the cerebellum accompanied by impaired oxidative phosphorylation. To investigate the molecular mechanisms of FRDA disease and therapy, a number of different FRDA models have been implemented[17]. Modeling of the human FXN trinucleotide expansion is technically complex but it was achieved by stable integration of a human BAC or YAC containing the FXN with the expanded GAA sequence with the simultaneous silencing of the mouse ortholog[18,19]. However, these mutant mice generally display only mild neurological and peripheral symptoms lacking overt neurodegeneration as displayed by the human pathology. Thus, implementing disease modeling with human cells is of particular importance for this disorder. Primary fibroblasts and lymphocytes from FRDA patients are easily accessible, but they did not spontaneously exhibit the cardinal disease hallmarks, while neurons and cardiomyocytes, the cell types particularly affected in the disease, cannot be obtained directly from patients. Hence, stem cell reprogramming technology offers a convenient method to generate disease-specific induced pluripotent stem cells (iPSCs) acting as a long-term source of vulnerable somatic cells in FRDA. Interestingly, FRDA iPSCs showed enduring epigenetic silencing of the FXN gene locus, low frataxin protein levels with GAA repeat instability in culture[20,21]. FRDA iPSCs showed unaltered ability to differentiate into neuronal cells that exhibited some delay in functional maturation, decrease in mitochondrial membrane potential, low levels of Iron–sulfur (Fe–S) containing proteins and heightened antioxidant response[22,23]. The mild phenotype of FRDA iPSC-derived neurons may result from the heterogeneity of the cell population, with the majority of central neurons known to be less sensitive to reduced levels of frataxin than peripheral sensory neurons. Hence, generation of FRDA peripheral sensory neurons, ideally in the spatial context of the DRG structure, could present an elegant experimental system to further evaluate biochemical and cellular alterations triggered by FXN silencing.

At present, there is no effective treatment for FRDA which can either ameliorate the symptoms or modify its pathological progression[24,25]. To date, several gene therapies have been proposed and are currently in development. For example, selected HDAC inhibitors were shown to reactivate FXN expression by promoting histone acetylation in different FRDA cellular and animal models[26,27]. However, clinical implementation of these drugs awaits formal assessment of their efficacy and safety profiles in a large cohort of patients. Gene replacement therapy for FRDA is also under development and has shown initial efficacy in reversing the cardiomyopathy in FXN conditional mutant mice[28]. In principle, the use of engineered nucleases could enable the reactivation of the silenced endogenous FXN gene. Indeed, promising results in this direction have been reported using TALEN and CRISPR/Cas9 nucleases to excise the expanded GAA repeat sequence, and engineered nucleases fused to a transcription activator effector have been used to force the expression of the silenced FXN gene[29–31]. However, due to the lack of robust FRDA models these strategies have yet to demonstrate their ability to recover frataxin levels. Thus, whether these treatments are powerful enough to rescue the mitochondrial and cellular pathological phenotypes of FRDA remains yet to be determined.

In this study, we established a patterned iPSC-derived sensory neuronal circuitry between DRG organoid (DRGO) sensory neurons and muscle intrafusal fibers for an improved in vitro modeling of the FRDA pathology. Using FRDA iPSCs with different pathogenic GAA expansions, we defined the neuronal impairment in survival, axonal morphology, mitochondria mass, and neuronal-muscle synapses upon FXN silencing. Further, we have established a correlation between the rescue of pathological defects and epigenetic changes upon the trinucleotide expansion removal with the difference in the size of two genomic deletions in FXN intron 1 by CRISPR/Cas9 targeted engineering.

## Results

### CRISPR/Cas9-mediated excision of the FXN intronic GAA expansion in FRDA iPSCs.

To establish a stem cell model of FRDA, we generated transgene-free iPSCs from fibroblasts of two patients, PTL and PTS, with different symptomatic disease severity and carrying either 460/930 or 330/530 repeats, respectively (Fig. 1a and Supplementary Fig. 1a, b). PTL presented with an early onset disease at age 13, and at 36 exhibited marked scoliosis, saccadic slowing, modest dysarthria, loss of walking and upright position, foot plegia, distal hyposthenia of the lower and upper limbs, proximal stenia of the upper limbs and apallesthesia in the lower limbs. PTS developed a late onset disease at age 32, and at 52 presented mild dysarthria, horizontal nystagmus, mild dysmetria and involuntary tremor, ataxic gate and no heart disease. The disease onset and pathological conditions in these two patients match the overall clinical progression described in previous studies indicating that the GAA expansion size in FXN intron 1 well correlates with onset and severity of FRDA[6]. iPSC

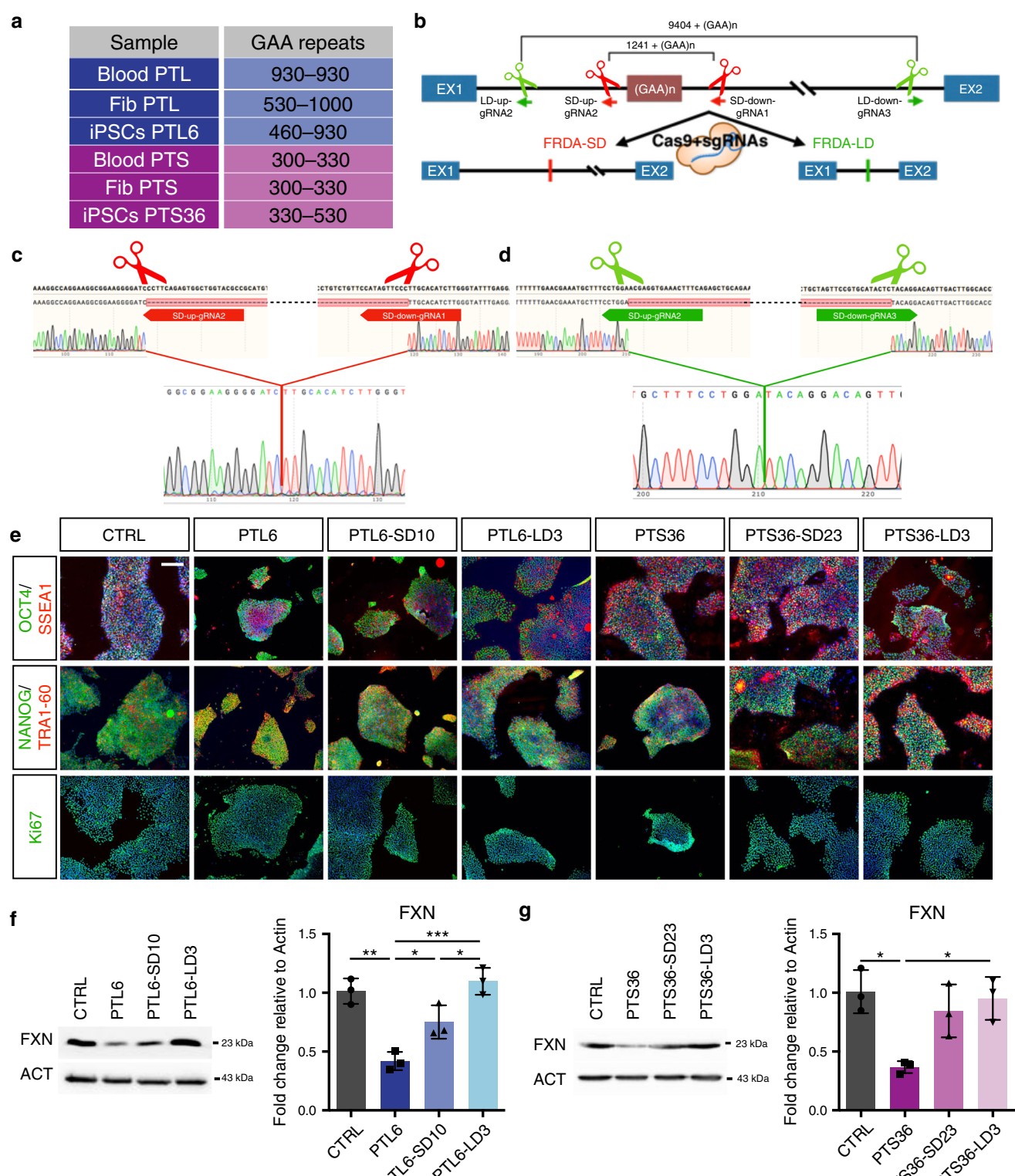

**Fig. 1 Generation of FRDA patient and isogenic CRISPR/Cas9 targeted iPSCs. a** Annotation of GAA repeats in different cell lines from FRDA patients PTS and PTL. **b** Illustration of CRISPR-based deletions in *FXN* intron 1. Pairs of sgRNAs drive Cas9-based targeted excision to proximal sites (red) flanking the GAA tract and long distal sites (green). Thus, short (SD) or long deletions (LD) of the *FXN* intron 1 are generated in FRDA-SD and FRDA-LD iPSC lines, respectively. **c, d** Representative Sanger sequencing confirming the generation of the short (**c**) and long deletion (**d**) in FRDA patient iPSCs. **e** Immunocytochemistry of the crucial pluripotency markers OCT4, NANOG, SSEA1 and TRA-1-60 on CTRL and untreated or targeted patient iPSCs. Scale bar, 100 μm. **f, g** Quantitative analysis of frataxin protein levels in PTL6 (**f**) and PTS36 (**g**) and their relative targeted iPSC lines. Protein levels are normalized to Actin. Mean ± s.d., $n = 3$ independent experiments, 24–36 organoids/line/experiment. *$P < 0.05$; **$P < 0.01$; ***$P < 0.001$; one-way ANOVA with Bonferroni correction.

lines for both patients were stabilized and selected based on their gain of expression of several crucial pluripotency markers and multi-lineage cell differentiation competence (Fig. 1e, Supplementary Fig. 2b, c). We, then, selected the iPSC lines PTL6 and PTS36, one for each patient. As already reported, we noted different independent events of contraction or expansion of the GAA tract among primary blood cells and fibroblasts of the patients and the reprogrammed iPSCs, confirming the high instability of this region (Fig. 1a and Supplementary Fig. 1a, b)[23]. Next, we sought to precisely delete the expanded repeats using CRISPR/Cas9 technology to generate two different deletions in either cell line, by either removing only the repeats (short deletion, SD) or excising almost the entirety of the long FXN intron 1 that contains the GAA expansion (long deletion, LD). For the short deletion, gRNAs were designed flanking the 5′ and 3′ regions of the GAA tracts (SD-up-gRNA and SD-down-gRNA, respectively). For the long deletion, gRNAs were designed close to the boundaries of intron 1 (LD-up-gRNA and LD-down-gRNA), about 1000 bp downstream to the 5′ and 500 bp upstream to the 3′ intron/exon junction, in order to leave splicing regulatory regions fully untouched (Fig. 1b and Supplementary Fig. 1c). Three different gRNAs for each intronic region were assessed for their INDEL rate activity using the T7 Endonuclease I assay. With this analysis, we were able to assemble two pairs of sgRNAs with the highest deletion activity (SD-up-gRNA2/SD-down-gRNA1 and LD-up-gRNA2/LD-down-gRNA3) for excising the triplet repeat expansion with a short and long deletion (Supplementary Fig. 1d, e, arrows). Then, PTL6 and PTS36 iPSCs were lipofected with plasmids expressing either sgRNA pair together with spCas9 and subsequently expanded into single clones, each of which was tested for the presence of the desired genome modifications. This molecular analysis identified several clones with deletions in one or both alleles of FXN intron 1 from which we chose, for each iPSC line, one clone with the short and a second clone with the long genomic deletion in homozygosis. Clones SD10 and LD3 for PTL6 iPSCs and SD23 and LD3 for PTS36 iPSCs were selected as they exhibited the desired modification as ascertained by direct sequencing of the genomic junctions outside the deleted region (Fig. 1c, d and Supplementary Fig. 1a, f, g) and did not present any off-target effects within the top four gRNA homology sites (Supplementary Fig. 1h–k). Moreover, all four targeted patient iPSC lines showed correct colony morphology, normal karyotype and expression of crucial pluripotency markers, as well as in vitro multi-lineage cell differentiation competence and teratoma potential (Fig. 1e and Supplementary Fig. 2a–c).

In FRDA, the length of the GAA expansion correlates proportionally with the extent of the FXN gene silencing[6]. Accordingly, patient iPSCs showed reduced levels of both transcripts and protein respect to unrelated control iPSCs (CTRL) derived from an unaffected individual with a physiological triplet number (Fig. 1f, g, and Supplementary Fig. 2d, e). Interestingly, both targeted PTS36-SD23 and -LD3 lines exhibited an extensive recovery in frataxin expression. On the contrary, in PTL6 iPSCs—those with the longer repeat expansions—only the targeted LD3 clone, but not the SD10, showed a significant recovery of frataxin expression (Fig. 1f, g, and Supplementary Fig. 2d, e). These results indicate that deleting exclusively the repeat expansion is not always sufficient to fully rescue frataxin silencing—in particularly in those cells carrying a long tract of repeats in FXN intron 1.

**In vitro generation and characterization of iPSC-derived DRGOs.** Peripheral neurons in DRGs and large projection neurons in the cerebellar deep nuclei are the primarily and most aggressively affected in FRDA[1,32]. Accordingly, bioinformatic analysis of publicly available RNA-seq datasets showed that FXN expression is higher in DRGs and cerebellum respect to other brain regions as previously suggested (Supplementary Fig. 3a)[7]. Moreover, we found that FXN expression levels doubled in peripheral sensory neurons obtained from iPSCs with a 2D-differentiation protocol (iPSC-SNs) when compared with other iPSC-derived neuronal subtypes, such as forebrain GABAergic (GABA-iNs) and cortical excitatory neurons (iNs) (Supplementary Fig. 3b)[33,34]. Thus, FXN expression modulation might reflect alternative mitochondrial metabolic states in different neuronal subtypes and its transcriptional expression pattern is recapitulated using iPSC-derived neuronal derivatives. Hence, it is of high relevance to study the effects of frataxin silencing in iPSC-derived DRG sensory neurons to understand their specific vulnerability in FRDA.

2D-differentiation strategies have been employed to generate sensory neurons from iPSCs, but these do not form DRG-like structures. It is reported that 2D iPSC-SNs form unorganized clusters in a random pattern with unproven concomitant participation of different neuron subtypes and glial cells, fail to recapitulate the DRGs spatial architecture and cellular diversity[35,36]. Of note, even if some recent studies showed new derivation of FRDA SNs by direct cell reprogramming or iPSC differentiation[37,38], when we tried to generate SNs from FRDA patient iPSCs, an extensive cell death occurred which prevented obtaining neuronal cultures with sufficient numbers of PRPH+/BRN3A+ sensory neurons (Supplementary Fig. 3c). We hypothesized that the absence of glial cells and sufficient cell-to-cell contacts in 2D cultures may further reduce viability of FRDA iPSC-SNs. To generate 3D DRG-like structures, single iPSC aggregates were cultured in low-adhesion 96-well plates for 16 days in vitro (DIV) and exposed to a sequential combination of small molecules previously showed to differentiate iPSCs into neural crest neuronal derivatives (Fig. 2a)[35]. Neuralization of iPSC aggregates was initially induced with LDN-193189/SB431542 for 10 days, followed by supplementation with the three inhibitors SU5402, CHIR99021 and DAPT, known to promote neuronal maturation and cell-cycle exit[35]. At day 10, small molecules were withdrawn, and the maturation medium was added containing a cocktail of four different neurotrophins and ascorbic acid. At DIV 16, neural aggregates were transferred into Matrigel-coated dishes to promote their adhesion to the plate in N2 medium containing the maturation factors. In this condition, soon after adhesion neuronal aggregates started to emit a radial array of axonal projections that spread over the plate reaching a long distance from the central cell clusters (Fig. 2b). To eliminate the residual proliferation of remaining neural progenitors, aggregates were exposed to the anti-mitotic reagent 5-fluoro-2-deoxy-uridine (FUDR) between days 17–26. At DIV 40, most of the neural aggregates showed the generation of a large set of radial projections decorated with peripherin (PRPH), a type III intermediate filament primarily found in peripheral neurons (Fig. 2b). During neuronal differentiation, we observed two distinct but consecutive waves of NEUROG1 and NEUROG2 gene expression (Supplementary Fig. 3d), resembling what happens in DRG embryogenesis in vivo during the consecutive generation of firstly the NTRK3+/NTRK2+ and secondly the NTRK1+ neurons[39,40]. To further confirm the peripheral sensory identity of the neurons, we proceeded with candidate gene expression analysis, revealing a strong upregulation of several peripheral sensory neuronal markers, including POU4F1, GFRA2, P2RX3, PTPRT, NRG1, vGLUT1, and sodium channel genes (Supplementary Fig. 3e)[41]. Likewise, βIII-Tubulin, PRPH, and NF200, a type IV intermediate filament associated with high caliber neurites, were clearly observed within the axonal projections (Supplementary Figs. 2c and 3f). In addition, nuclear

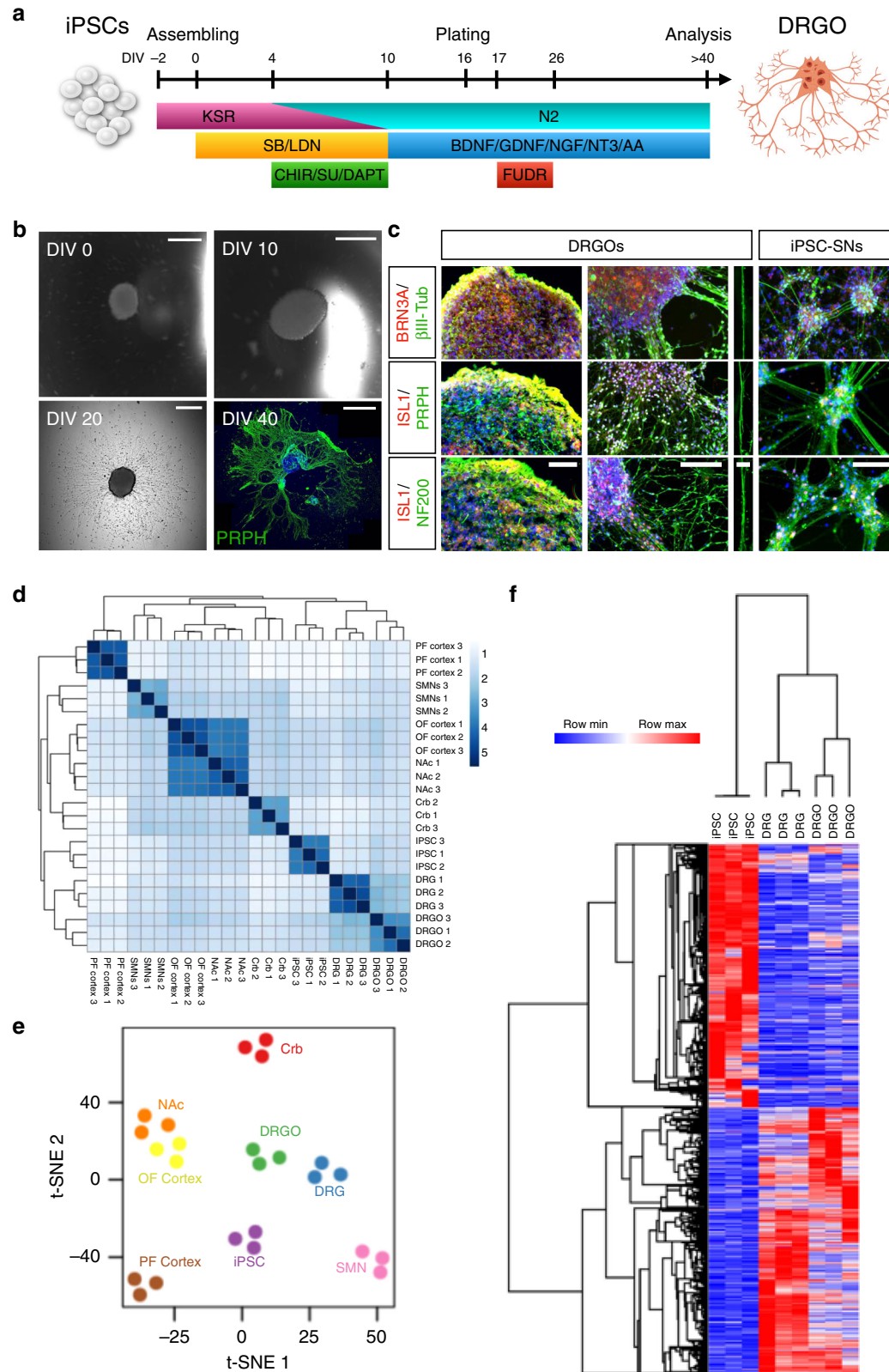

transcription factors BRN3A, ISLET1 were readily detectable by immunostaining in a large fraction of the cells within the aggregates (Fig. 2c and Supplementary Fig. 3g). Thus, morphology, axonal projections and key molecular markers of iPSC-derived neuronal aggregates are very reminiscent of primary DRGs isolated from rodents and cultured in vitro[42]. Hence, we named these structures DRGOs. Interestingly, both comparative

immunohistochemistry and expression analyses confirmed that differentiation in DRGOs sustained a small, but consistent, increase in peripheral sensory neurons on the total amount of cells with respect to cultures differentiated with a 2D protocol (Fig. 2c and Supplementary Fig. 3e–g).

To further interrogate the transcriptome of iPSC-derived DRGOs and compare it with that of primary DRGs, we carried

**Fig. 2 Generation and characterization of dorsal root ganglia organoids (DRGOs). a** Illustration of the 3D culture system and sequential exposure to small molecules over time to obtain DRGOs. **b** Images of key steps in DRGO generation: DIV 0, cell aggregate assembling; DIV 10, neuralized cell aggregates; DIV 20, generation of a star-like web of axonal projections around the central mass; DIV 40, axonal projections positive for the peripheral marker peripherin (PRPH). Scale bars, 1 mm (DIV 40), 500 μm (DIV 0–20). **c** Immunocytochemistry in DRGOs and 2D-differentiated peripheral neurons (iPSC-SNs) at DIV 40 for proteins localized along neuronal projections (ßIII-Tubulin, PRPH, NF200) and the sensory neuron-specific transcription factors BRN3A and ISL1. Scale bars, 100μm (first column), 50μm (second and fourth columns), 5 μm (third column). **d** Correlation Heatmap showing absolute distances (Pearson $R^2-1$) between transcriptomes of different brain regions and primary neuronal subtypes: PF cortex: prefrontal cortex; OF cortex: orbitofrontal cortex; Crb: Cerebellum; NAc; Nucleus accumbens; MSNs: motor spinal neurons; DRG; Dorsal Root Ganglia; DRGOs; in vitro differentiated iPSC-derived DRGs. The correlation between samples is also shown as an unsupervised hierarchical clustered dendrogram on the sides. **e** Whole-transcriptome analysis using the t-Distributed Stochastic Neighbor Embedding (t-SNE) analysis with a view of the sample distribution along the first two dimensions. **f** Gene expression heatmap showing the genes differentially expressed either in DRGs vs iPSC, or DRGO vs iPSC; the correlation between samples is also shown as an unsupervised hierarchical clustered dendrogram on the side.

out RNA-seq profiling of DRGOs at day 40 and collate it with similar publicly available datasets from native DRGs. Overall, the transcription profiles of iPSC-derived DRGOs and human DRGs displayed the strongest correlation, when compared to similar datasets of other primary neuronal subtypes, brain regions and iPSCs, as summarized by the unsupervised hierarchical clustering (UHC) of whole transcriptome Pearson correlation coefficients (Fig. 2d), t-SNE analysis (Fig. 2e), and UHC of those genes being differentially expressed in both DRGs or DRGOs versus iPSCs (Fig. 2f). In particular, DRGOs showed a general and consistent silencing of genes specifically enriched in undifferentiated iPSCs coupled with a significant upregulation of genes involved in sensory neuronal specification (Fig. 2f). Furthermore, GO analysis of these genes revealed a positive functional enrichment in DRGOs of multiple biological processes related to peripheral sensory neuron biology, including glutamatergic neuronal and synaptic identity, neuronal activity and neuromuscular connectivity (Supplementary Fig. 4a, top). On the contrary, GO gene sets related to stem cell maintenance and proliferation were negatively enriched in DRGOs (Supplementary Fig. 4a, top). Conversely, some transcriptional differences were also apparent between primary DRGs and DRGOs, that might be mostly explained by the presence in primary DRG samples of other cell types of the vasculature, blood and immune system (Supplementary Fig. 4a, bottom). More importantly, DRGOs and primary DRGs clustered together when considering only those genes known to be pivotal for sensory neuronal development and sensory perception, thus illustrating the similarities between the somatic and stem cell-derived DRGs (Supplementary Fig. 4b, c). We then asked how DRGOs resemble primary sensory neurons when compared to previously published 2D-based protocols to generate sensory neurons through iPSC differentiation (iPSC-SNs)[35] or direct conversion of somatic fibroblasts (iSNs)[38,43]. Gene set functional enrichment analysis of the genes significantly up- or down-regulated in the somatic DRGs versus undifferentiated iPSCs (Supplementary Fig. 4d) and UHC of the genes associated to peripheral sensory neuronal development and functioning (Supplementary Fig. 4e) clearly showed that all three methodologies generated sensory neurons strongly resembling, at the transcriptomic level, those of primary DRGs. Thus, these results revealed that similar to 2D-differentiated neurons, DIV 40 DRGOs exhibited a global transcriptional reconfiguration towards sensory neuronal cells and supporting peripheral glial cells. Furthermore, they better recapitulate the spatial organization and cellular composition respect to classical 2D-differentiated neuronal cultures.

**Dissecting DRGOs cellular composition by immunohistochemistry and single-cell transcriptomic profile**. Initially, in order to determine the presence and identity of the sensory neurons, we carried out immunostainings with specific molecular

markers on DIV 40 DRGOs. Though with some variability between independent experimental batches, we confirmed the presence of TRKA$^+$/TRKC$^-$ nociceptors (9.6% ± 1.8%), TRKB$^+$/TRKC$^+$ mechanoreceptors (32.5% ± 3.6%) and TRKB$^-$/TRKC$^+$ proprioceptors (31.1% ± 8.4%) (Fig. 3a, b and Supplementary Fig. 5a, b). We also observed NF$^+$ neurons immunodecorated with other different combinations of TRK receptors (such as TRKA$^+$/TRKB$^+$ neurons; Fig. 3a, arrowhead) and likely representing neuronal precursors lacking a specific subtype identity[44]. Consistently, we also detected a fraction of p75$^+$ immature precursors cells (9.1% ± 1.4%) (Supplementary Fig. 5e, f). Finally, we also detected the presence of glial cells, such as S100$^+$ satellite cells (10.8% ± 2.4%) (Supplementary Fig. 5c, d). Bulk RNA-seq analysis confirmed that DRGOs and primary DRGs are closely related with regard to the expression of gene sets that characterize nociceptive and mechanoreceptive neurons as well as satellite cells and Schwann cells, thus corroborating the similarities between the somatic and stem cell-derived DRGs (Supplementary Fig. 5g, h, k)[45,46]. On the other hand, DIV 40 DRGOs appeared immature compared to their primary counterparts as demonstrated by the expression of neuronal precursor specific genes such as NEUROG1, NEUROG2, PAX6, and SOX2 (Supplementary Fig. 5j), as well as the relatively lower levels of some proprioceptor (SPP1, RUNX3, NEFH, and PVALB) and nociceptor (NTRK1, TRPV1, and MRGPRD) specific transcripts (Supplementary Fig. 5g, i).

To better resolve the cellular diversity of DRGOs, we performed scRNA-seq on 5,363 single cells isolated from two independent batches of three DIV80 DRGOs. Unsupervised UMAP analysis identified 14 cell clusters based on their overall gene signature (Fig. 3c–j). Among the eight neuronal cell clusters (C1–C8, 73.8% of total cells), C1 was composed of NEUROG1$^+$/NEUROG2$^+$ neuronal progenitors lacking a yet defined sensory neuronal subtype identity (20.4% of the total neuronal cells). C2 and C3 were mainly composed of putative proprioceptive neurons with C2 cells still expressing NEUROG2, indicating an immature state, while C3 including a more mature cell population expressing NTRK3 and MAP2, but not NEUROG2 and NTRK2 (29.2% of total neuronal cells). Nociceptors were clustered in C4, defined mainly by the selective expression of NTRK1 together with the strong co-expression of SST and TH (3.9% of the total neuronal cells) (Fig. 3c–j). C5, C6, C7, and C8 were related to putative mechanoreceptors and their precursors (46.5% of the total neuronal cells). In particular, C5-C7 distinguished immature mechanoreceptors co-expressing NEUROG1/2 with NTRK2/3, while C8 included MAP2$^+$/NEUROG1$^-$/2$^-$ mature neurons. Overall, five out of eight neuronal clusters represented precursor neuronal cells. Moreover, few NTRK1 expressing cells were present in C4 indicating few fully mature nociceptors. Intriguingly, five out of the fourteen clusters are related to supporting cells, including satellite cells expressing SOSTDC1 and MSX1

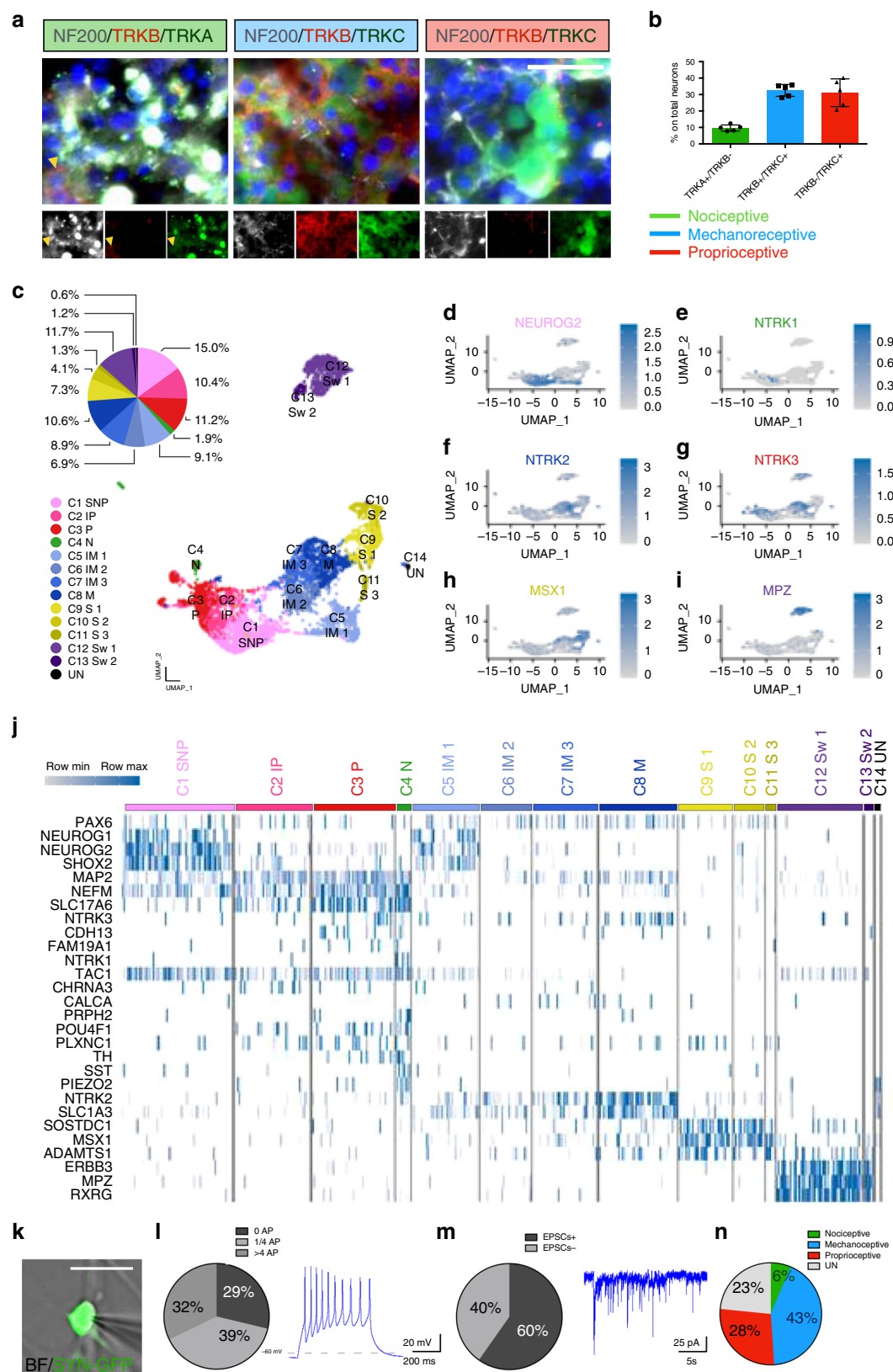

(C9–C11, 12.6% of total cells) and *MPZ⁺/SOX10⁺* Schwann cells (C12-13, 12.9% of total cells). Altogether, single-cell transcriptomic enabled to define the detailed cellular composition of the DRGOs revealing a complex variety of mature neurons and their precursors which are normally found during development of native DRGs.

Next, we moved to assess the electrophysiological properties of DIV 80 DRGO sensory neurons. Patch-clamp recordings were performed on DRGOs previously transduced with a synapsin-GFP lentiviral vector for selective visualization of neurons in real-time (Fig. 3k). Roughly 70% of the GFP⁺ neurons reached functional maturation showing hyperpolarized membrane

**Fig. 3 Single-cell transcriptomic profiling of DRGO cellular subtypes. a, b** Immunocytochemistry on DIV 40 DRGOs (**a**) and relative quantification (**b**) of TRK A/B/C combination distinguishing the nociceptive, mechanoreceptive, proprioceptive and undefined neuronal subtypes (arrowhead). Mean ± s.d. $n =$ 5 independent experiments, 6-9 organoids/experiment. Scale bar, 10 µm. **c** Uniform Manifold Approximation and Projection (UMAP) plot displaying multidimensional reduction and clustering of single-cell RNA-Seq data from DIV 80 DRGOs, and pie chart showing relative abundances of the different cell types. SNP, sensory neuron precursors; IP, immature proprioceptors; P, proprioceptors; N, nociceptors; IM, immature mechanoceptors; M, mechanoceptors; S, satellite cells; Sw, Schwann cells; UN, unknown. (**d–i**) UMAP plots highlighting normalized expression values of NEUROG2 associated to immature sensory neurons (**d**), NTRK1 associated to immature and mature nociceptors (**e**), NTRK2 associated to mechanoreceptors (**f**), NTRK3 associated to both proprioceptors and mechanoreceptors (**g**), MSX1 associated to satellite cells (**h**), and MPZ associated to Schwann cells (**i**). **j** Heatmap showing normalized expression values of cell lineage-specific genes within the different clusters. **k** Representative image of a Syn-GFP patched neuron. Scale bar, 50 µm. **l** Percentage of DRGO neurons with different action potential patterns following a 100 pA current injection and recorded by current-clamp electrophysiological registrations. $n = 3$ independent experiments, 2 DRGO/experiment. **m** Percentage of DRGO neurons that exhibit spontaneous excitatory postsynaptic currents in voltage-clamp electrophysiological recording. $n = 3$ independent experiments, 2 DRGO/experiment. **n** Percentage of patched DRGO neurons that exhibit specific neuronal subtypes after single-cell-RT-PCR. $n = 3$ independent experiments, 2 DRGO/experiment.

potential and the ability to spike repetitive action potentials (Vrest of $-53.8 \pm 10.3$ mV, $n = 87$ neurons) (Fig. 3l). Further, characterization of the synaptic activity revealed the presence of spontaneous excitatory postsynaptic currents (EPSCs) in ~60% of the analyzed neurons (Fig. 3m), indicating the formation of a functional synaptic network. Soon after recording, the soma of the neurons were isolated to perform gene expression analysis for neuronal subtype identification. Molecular profiling revealed that all three classes of sensory neurons were recorded although at different frequency (6% NTRK1$^+$ nociceptors; 43% NTRK2$^+$/3$^+$ mechanoceptors; 28% NTRK2$^+$/3$^-$ proprioceptive neurons; 2% unidentified cells) (Fig. 3n). The results revealed that DRGO differentiation enabled the specification and functional maturation of the three main sensory neuronal subtypes.

**Generation of the proprioception sensory pathway between DRGO sensory neurons and muscle fibers.** DRGOs have a stereotypic organization in the dish with all the soma clustered together and axons extending radially outwards from the soma forming a star-like network of projections. We anticipated that this geometry offered an ordered system that would facilitate evaluating the formation of connections with other cell elements in the culture. Thus, we sought to establish ordered connectivity between DRGO sensory neurons and muscle fibers to recapitulate the proprioception sensory pathway in vitro. In particular, we aimed to reconstitute the muscle spindle receptor which is the sensory organ responsive to changes in skeletal muscle length and involved in proprioception and the acquisition of body spatial information[47,48]. The muscle spindle receptor is constituted by a particular multinucleated muscle fiber, known as an intrafusal muscle fiber, which is enwrapped multiple times by the axonal terminals of large proprioceptive neurons with an annulospiral geometry[49]. To generate muscle spindles, the intrafusal muscle fibers were first generated from human myoblasts after treatment with NRG1, which simulates axonal contact and cues maturation into intrafusal fibers (Supplementary Fig. 6a)[50,51]. At DIV 18, 40-60% of multinucleated muscle fibers could be classified as intrafusal subtypes, positive for slow developmental Myosin heavy chain (sdMyHC, S46 antibody) and Myosin heavy chain 2 (MyHC2, A4,74 antibody) (Supplementary Fig. 6b–d). Conversely, at DIV 30 we observed a strong reduction of sdMyHC+ and MyHC2+ intrafusal fibers, due to the absence of neuronal contact, necessary for their long-term survival in culture, as previously reported (Supplementary Fig. 6b–d)[52].

With these premises, to replicate the sensory portion of the muscle spindle proprioceptive unit, we seeded DIV 16 DRGOs on intrafusal fibers in a window between 6 and 13 DIV after NRG1 addition, to allow proper DRGO axonal spreading, neuromuscular connection and to increase the survival of intrafusal fibers over a longer period of time. Confocal microscopy, as well as

electron microscopy, at 7 DIV of co-culture clearly showed the characteristic annulospiral wrapping of neuronal endings around the intrafusal muscle fibers (Fig. 4a–c and Supplementary Movie 1). Interestingly, intrafusal fibers were able to survive as muscle spindles after 50 DIV when in co-culture—a condition that could not be achieved in the absence of DRGOs. At this time, clusters of the synaptic proteins Calretinin (CALB2) and vGLUT1 within the NF200+ neuronal annulospiral endings reflected muscle spindle maturation (Fig. 4d–k) and pointed to the formation of neuromuscular synaptic structures. Moreover, clathrin-coated vesicle formation is observed at this time point, which may indicate synaptic-like vesicle recycling (Fig. 4l). These results strongly suggest that DRGO sensory neurons are able to autonomously develop elaborated connections targeting peripheral cells to generate complex structures such as muscle spindles, thus, supporting the paradigm of in vitro self-organization of complex organ-like structures including different cell types.

Maturation of DIV 50 DRGO sensory neurons maintained in co-culture with intrafusal fibers, was confirmed by the strong expression of mature neuronal genes encoding for the regulator of myelination, *NRG1*, and main sodium ion channels *SCN3A*, *SCN8A*, and *SCN9A* (Supplementary Fig. 6e–h). In addition, we scored the presence of diffuse vGLUT1 and Synapsin (SYN) positive puncta along the axonal projections, suggesting an active machinery for neurotransmitter trafficking (Fig. 4m, n).

In summary, we defined the conditions to establish a stable and patterned connection between DRGO sensory neurons and muscle cells for the generation of well-structured muscle spindle receptors in vitro that provide a robust cellular platform to investigate FRDA pathophysiological mechanisms.

**Excision of *FXN* intron 1 is mostly effective in reactivating *FXN* expression in patient DRGOs.** Next, we generated DRGOs from the PTL6 and PTS36 patient iPSCs as well as CRISPR-deleted derivatives to evaluate frataxin levels before and after targeted excision of the GAA tracts. As expected, FRDA DRGOs showed a strong downregulation of *FXN* mRNA and protein levels (Fig. 5a–d). Interestingly, similar to what was described earlier in undifferentiated iPSCs, *FXN* silencing was stronger in PTL6 with respect to PTS36 DRGOs. Conversely, frataxin silencing at protein level was evidently stronger in PTL6 respect to PTS36 DRGOs, thus, indicating that DRGOs better recapitulate FRDA molecular effects than iPSCs. Moreover, although an extensive rescue of frataxin levels was observed in DRGOs from both FRDA-LD lines. In contrast, DRGOs from patient cells with a short deletion showed only a mild increase in frataxin mRNA and protein as compared to the parental lines, never fully reaching the control DRGO levels (Fig. 5a–d). These results strongly suggested that the additional removal of the surrounding

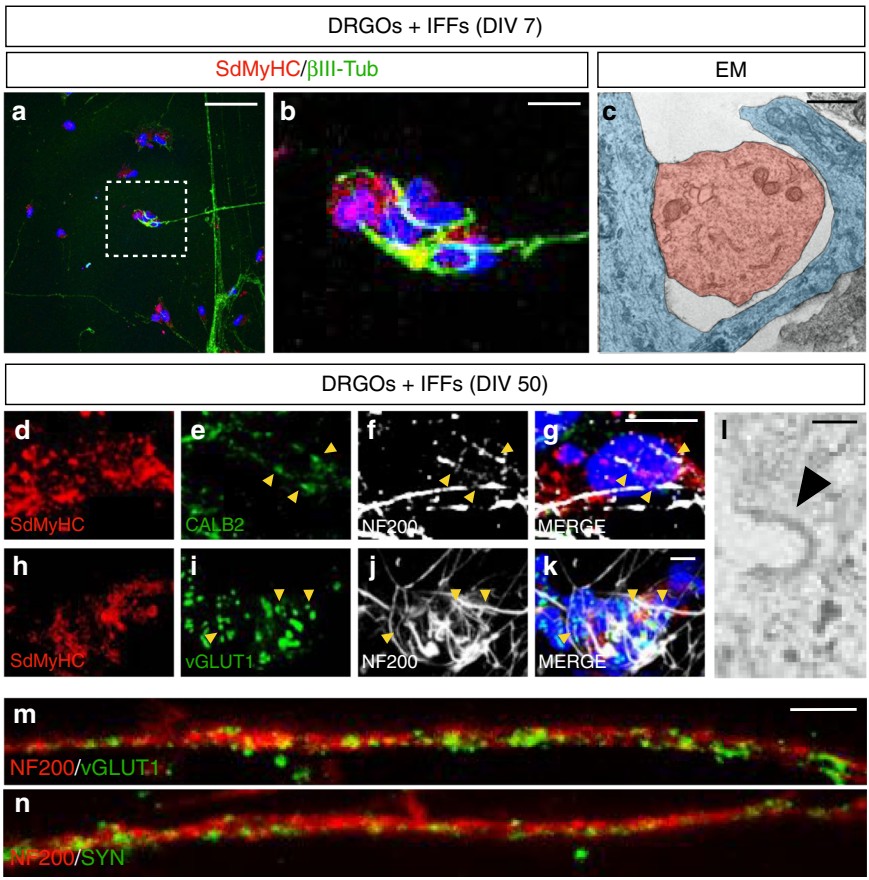

**Fig. 4 In vitro reconstitution of the muscle spindles. a** Co-cultures of DRGOs and intrafusal muscle fibers (IFFs) at DIV 7 with single axonal projections contacting IFFs visualized by immunofluorescence staining for the intrafusal muscle fiber marker S46 (red), and the neuronal protein ßIII-Tubulin (green). Scale bar, 50 μm. **b** Enlarged view of the boxed area in **a** reveals the annulospiral wrapping of the axonal terminal around the multinucleated intrafusal muscle fiber within the central domain where nuclei are concentrated (blue, nuclear staining with Hoechst). Scale bar, 10 μm. **c** Electron microscopy (EM) imaging showing an axonal process (blue) which is wrapping a segment of the intrafusal muscle fiber (red). Scale bar, 1 μm. **d–k** Calretinin (CALB2) (**d–g**) and vGLUT1 (**h–k**) staining (arrowheads) along the NF200 + annulospiral axonal terminals of DRGOs in co-cultures with intrafusal muscle fibers (IFFs) stained with S46 at DIV 50. Scale bar, 10 μm. **l** Electron microscopy imaging showing an invaginating clathrin-coated vesicle (arrowhead) within the axonal terminal. Scale bar, 100 nm. **m, n** Immunofluorescence for the synaptic vesicle markers Synapsin and vGLUT1 (green) within axonal fascicles labeled by NF200 (red). Scale bar, 5 μm.

*FXN* intron 1 sequence might be required to obtain the most pronounced rescue of frataxin levels in DRGO cells.

Numerous studies have highlighted that, in FRDA cells, chromatin in *FXN* intron 1 exhibits high levels of repressive histone marks including H3K9m3 and increased loss of histone acetylation[9]. In addition, the increase of acetylated histones by treatment with HDAC inhibitors was found to promote *FXN* expression in FRDA cells[11,26,27]. In light of these results, we speculated that the full excision of the *FXN* intron 1 might remove additional repressed chromatin, which contributes to the repression of *FXN* transcription in diseased cells, and, therefore, further increases the reactivation of the gene. At the same time, we anticipated that the deletion restricted only to the GAA tract would not have an immediate impact on the repressed epigenetic state of the chromatin in regions immediately flanking the GAA tract or the adjacent *FXN* exon 1 sequences. To experimentally corroborate these assumptions, we profiled the levels of the epigenetic marks H3K9me3 and H3K9ac by ChIP-qPCR analysis, and DNA methylation by MeDIP-qPCR in DRGOs from unmodified and CRISPR-targeted patient iPSC lines (Fig. 5e–h). The H3K9me3 mark was found particularly enriched in all the regions surrounding the repeat expansion in PTL6 and less so in PTS36 patients with respect to control DRGOs, consistent with the strongest reduction in *FXN* expression in the former with

respect to the latter patient cell line (Fig. 5f). Similarly, the *FXN* promoter region was found to be strongly methylated particularly in PTL6 DRGOs (Fig. 5h). The H3K9ac mark was found to be substantially reduced in PTL6, but less in PTS36, with respect to control DRGOs, corroborating the chromatin differences between the two patient FRDA DRGOs (Fig. 5g). Thus, we focused on PTL6 cells and evaluated the epigenetic profiling after excision of the GAA tract only (PTL6-SD10) or included of the *FXN* intron 1 (PTL6-LD3). Interestingly, in PTL6-SD10 DRGOs, after removal of the only GAA expansion, the surrounding regions presented a minor reduction of H3K9me3 as compared to the parental line that, however, remained significantly higher with respect to the control counterpart (Fig. 5f). Thus, in PTL6-LD3 DRGOs, the additional excision of the *FXN* intron 1 granted the removal of a large region of repressed chromatin. In addition, the residual intronic regions of PTLS-LD3 DRGOs showed a marked reduction of H3K9me3 (Fig. 5f). We found a similar DNA methylation profile on the *FXN* promoter region, with no significant difference between control and PTL6-LD3 DRGOs (Fig. 5h). Furthermore, PTL6-SD10 and PTL6-LD3 cells displayed a partial restauration of H3K9ac mainly in the intronic region upstream the GAA expansion (Fig. 5g).

To explore the possibility that deletions might impact the splicing of the intronic region, we performed qPCR with primers

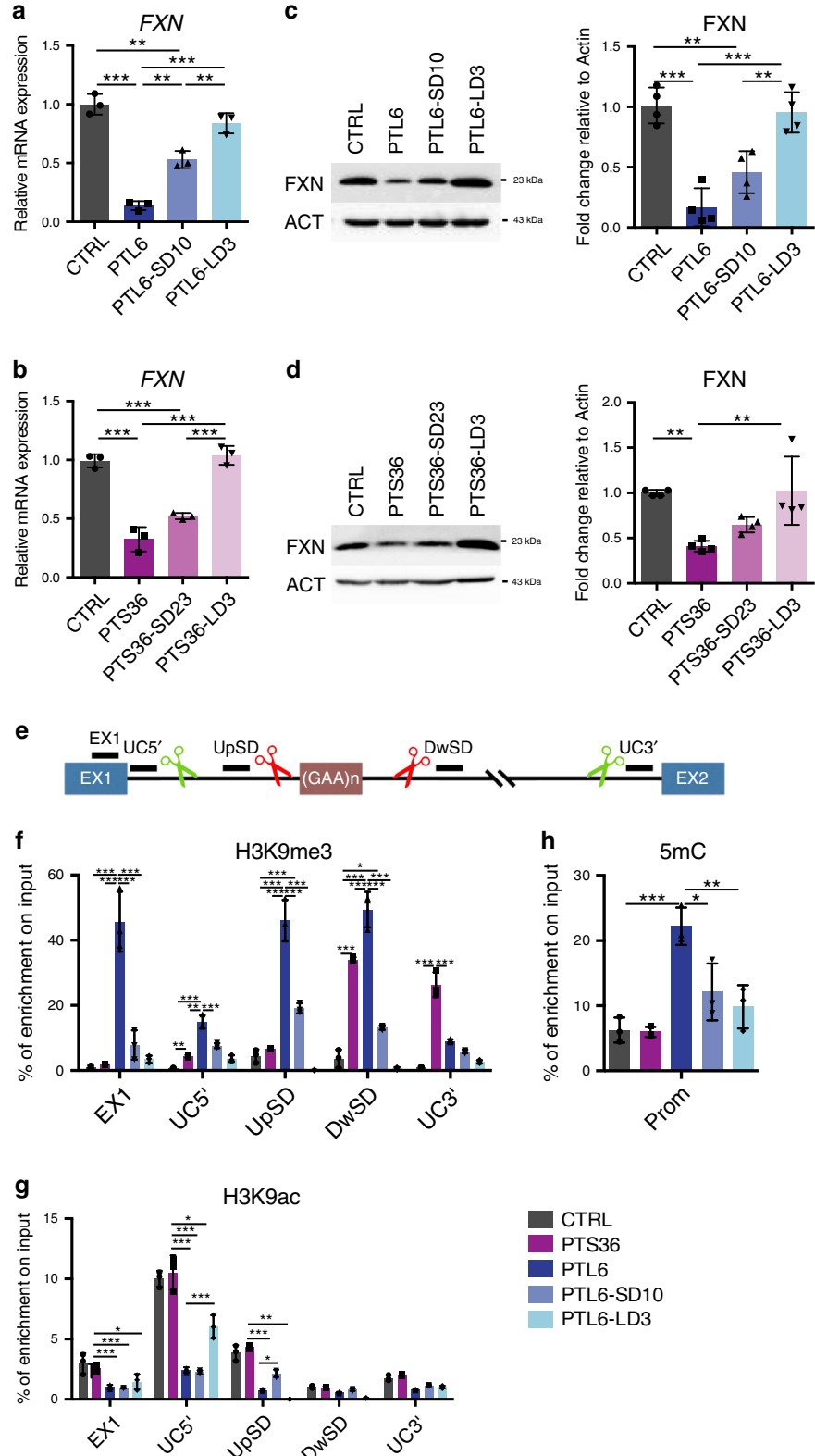

flanking the conjunction between *FXN* exon 1/2 junction site, but we did not detect unspliced forms (Supplementary Fig. 7a, b). Moreover, to ensure that removal of intron 1 did not disturb gene regulation in the *FXN* genomic locus, we evaluated the transcriptional expression of the *FXN* flanking genes. PIP5K1B and TJP2 gene expression were restored in LD3 respective to PTL6 parental DRGOs, while no alteration in FAM189A2

expression was detected between FRDA parental and LD DRGOs (Supplementary Fig. 7c–h).

**Defective survival and morphology of patient DRGOs are rescued by *FXN* intron 1 deletion**. Next, we generated DRGOs from patient iPSCs and their CRISPR-deleted derivatives to

**Fig. 5 Frataxin levels and epigenetic profiling in untreated and targeted patient iPSC-derived DRGOs. a**, **b** Quantitative analysis of *FXN* gene transcriptional levels in DIV 40 PTL6 (**a**), PTS36 (**b**) and their respective isogenic targeted DRGOs. Expression levels are normalized to actin. Mean ± s.d., $n = 3$ independent experiments, 8–12 organoids/line/experiment. **P < 0.01; ***P < 0.001; one-way ANOVA with Bonferroni correction. **c**, **d** Quantitative analysis of frataxin protein levels in control, PTL6 (**c**), PTS36 (**d**) and their respective isogenic targeted DRGOs. Protein levels are normalized to actin. Mean ± s.d., $n = 4$ independent experiments 24–36 organoids/line/experiment. **P < 0.01; ***P < 0.001; one-way ANOVA with Bonferroni correction. **e** Illustration of the genomic sites within the *FXN* exon 1 and its downstream intron where H3K9me3 relative abundance was profiled by ChIP-qPCR analysis. EX1, Exon 1; UC5′, Uncut region at 5′; UpSD, Region upstream Short deletion; DwSD, Region downstream Short deletion; UC3′, Uncut region at 3′. **f**, **g** Relative abundance of H3K9me3 (**f**) and H3K9ac (**g**) as measured by ChIP-qPCR analysis in different regions along the exon and intron 1 in FXN gene, in control, PTS36, PTL6 and its isogenic targeted DRGOs. Data are plotted as % of enrichment on the input for each sample. Mean ± s.d., $n = 3$ independent experiments 24–36 organoids/line/experiment. *P < 0.05; **P < 0.01; ***P < 0.001; two-way ANOVA. **h** qPCR analysis of independent meDIP samples, data show the level of DNA methylation of *FXN* gene promoter in control, PTS36, PTL6 and its isogenic targeted DRGOs. All data are plotted as % of enrichment on the input of each sample. Mean ± s.d., $n = 3$ independent experiments 24–36 organoids/line/experiment. *P < 0.05; **P < 0.01; ***P < 0.001; two-way ANOVA.

investigate the FRDA-associated cellular and molecular impairments. To interrogate these aspects of the disease in our model, we decided to focus on differentiated DRGOs of the PTL6 cell line, as it manifested the most prominent *FXN* downregulation while its targeted PTL6-LD3 subclone displayed the strongest *FXN* expression recovery. At DIV 16 during in vitro differentiation of iPSCs, we observed a marked decrease in PTL6 DRGO survival with respect to DRGOs generated with control cell lines (61.5% ± 5.3% and 95.8% ± 3.6%, respectively). In contrast, the fraction of seemingly healthy DRGOs was restored to control levels in both PTL6-SD10 and -LD3 DRGOs (96.9% ± 7.9% and 95.9% ± 3.3%, respectively). Similar results were found at DIV 40, when the elevated number of dying DRGO cells of patient cell lines compared to control DRGOs (27.4% ± 5.8% and 8.2% ± 3.4%, respectively) was extensively rescued in both PTL6-SD10 and PTL6-LD3 DRGOs (10 ± 3.7% and 9% ± 4.2%, respectively), as highlighted by cleaved Caspase 3 staining (Supplementary Fig. 8a–c). Next, we moved to analyze the morphological traits of DRGO generated from FRDA cell lines as compared to their isogenic -SD and -LD lines. Interestingly, FRDA DRGOs displayed a severe impairment in axonal spreading over the dish compared to control DRGOs (Fig. 6a–d). This axonal growth deficiency was only partially restored in -SD DRGOs, but almost fully rescued in -LD DRGOs (Fig. 6a–d). Interestingly, the PVALB molecular marker specific for proprioceptors was particularly affected in differentiating DRGOs, while the nociceptor-specific gene NTRK1 was only mildly impaired in differentiating patient organoids (Fig. 6e–h). Conversely, no change was detected in gene expression levels for the mechanoreceptor CACNA1H marker (Fig. 6i, j). Collectively, these results uncovered relevant disease-specific deficits in survival and morphology of DRGO neural cells particularly affecting the proprioceptive neurons that were partially recovered to different extents in targeted patient cells.

**FRDA deficits in Fe–S cluster protein biogenesis and antioxidant response are restored in PTL6-LD DRGOs.** Two of the main functions of frataxin include the biogenesis of Fe–S proteins and ROS control the disruption of which, especially in FRDA, has been shown to result in impaired mitochondrial biogenesis[23,53]. Thus, we measured mitochondrial DNA (mtDNA) copy number by comparing the relative levels of mitochondrial and genomic DNA (gDNA) and observed a decrease of mitochondrial biogenesis in PTL6 patient DRGOs (Supplementary Fig. 9a). Conversely, the total amount of mtDNA copies was completely restored in PTL6-LD3 DRGOs with the long genomic deletion (Supplementary Fig. 9a). FXN deficiency in FRDA is known to limit the cellular capacity to synthesize Fe–S clusters leading to the reduction of Fe–S proteins—such as Aconitase-2 (ACO2), a mitochondrial resident enzyme. The subsequent enhancement in

mitochondrial ROS triggers a compensatory response which upregulates antioxidant proteins, such as the cytoplasmatic and mitochondrial resident Superoxide dismutases (SOD1/2, respectively)[22]. Of note, patient DRGOs presented a marked upregulation of *SOD1/2* along with reduction of Aconitase-2 at both transcript and protein levels (Supplementary Fig. 9b–e). Thus, these results demonstrate that diseased DRGOs recapitulated many peculiar aspects of the pathological phenotype present in patient somatic cells. Furthermore, the response to oxidative stress in both the mitochondrial and cytoplasmic compartments, represented by the overexpression of SOD1 and SOD2, and the impairment in the generation of Fe–S protein biogenesis, as shown for ACO2, in patient parental DRGOs, were strongly rescued in isogenic PTL6-LD3 DRGOs. Finally, PTL6-LD3 DRGOs presented a reduction of ROS accumulation compared to their parental line (Supplementary Fig. 9f). Thus, impaired molecular pathways intrinsically associated with FRDA are reversed after reactivation of Frataxin in targeted patient DRGOs.

**Mitochondrial defects in patient DRGO neuronal axons is restored upon frataxin reactivation.** Next, we decided to fabricate a two-compartment microdevice in PDMS, where neuronal soma and axons of the DRGOs could be spatially and fluidically isolated to better examine the processes that take place in the two compartments[54–56]. The microfluidic chip was designed with two parallel lateral channels fed by reservoirs at the extremities to facilitate media changes (Fig. 7a, b). These channels were fluidically isolated except for the presence of array of long perpendicular microgrooves. Within the proximal lateral channel, we fabricated two adjacent chambers to house the soma of DRGO neurons while allowing the axonal projections to spread into the microgrooves and reach the distal lateral channel (Fig. 7a). To enhance axonal growth through the microgrooves, a gradient of the neurotrophins NT3 and BDNF between the two compartments was applied. This particular design of the microdevice offered us the possibility to focus our analysis selectively on the axonal projections protruding from DRGOs and extending through the microgrooves with single axon resolution. DIV 16 DRGOs were plated into the microfluidic device and two weeks later were stained for the mitochondrial protein TOMM20 together with the axonal marker neurofilament NF200 and analyzed by confocal microscopy. Remarkably, patient DRGO axons clearly presented a significant reduction in the total number of mitochondria. Moreover, remained mitochondria displayed a smaller and more circular morphology suggesting increased mitochondrial fragmentation (Fig. 7c–g and Supplementary Fig. 9g). On the contrary, PTL6-LD3 DRGO axons presented a number of mitochondria comparable with those counted in control DRGO projections with a rescue in size and circularity (Fig. 7c–g and Supplementary Fig. 9g).

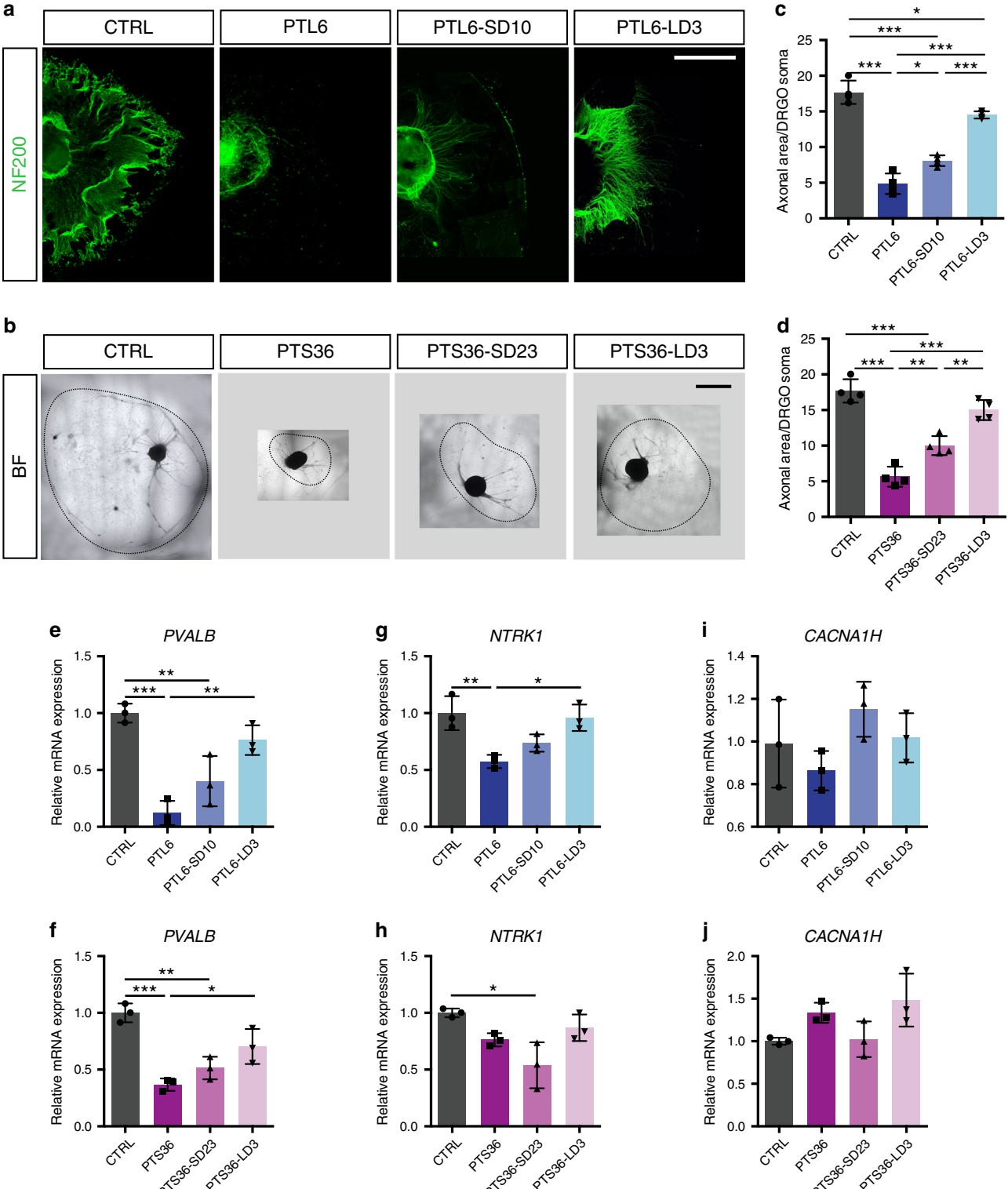

**Fig. 6 Disease phenotype amelioration in CRISPR/Cas9 edited FRDA-LD DRGOs. a, d** Representative images of FRDA patient line derived DRGOs along with short and long deletion isogenic lines, respectively, as compared with a control (CTRL) healthy donor derived DRGO stained for NF200 (**a**) and in BF (**b**), and neuritis area quantification for PTL6 and its isogenic lines (**c**) and PTS36 and its isogenic lines (**d**). Mean ± s.d., $n = 4$ independent experiments 3 organoids/line/experiment. *$P < 0.05$; **$P < 0.01$; ***$P < 0.001$; one-way ANOVA with Bonferroni correction. Scale bar: 500 μm. **e–j** Quantitative analysis of transcriptional levels of *PVALB* (proprirecptors) (**e**, **f**), *NTRK1* (nociceptors) (**g**, **h**) and *CACNA1H* (mechanoreceptors) (**i**, **j**) in control, FRDA patients and their isogenic lines. Expression levels are normalized to actin. Mean ± s.d., $n = 3$ independent experiments, 8–12 organoids/line/experiment. *$P < 0.05$; **$P < 0.01$; ***$P < 0.001$; one-way ANOVA with Bonferroni correction.

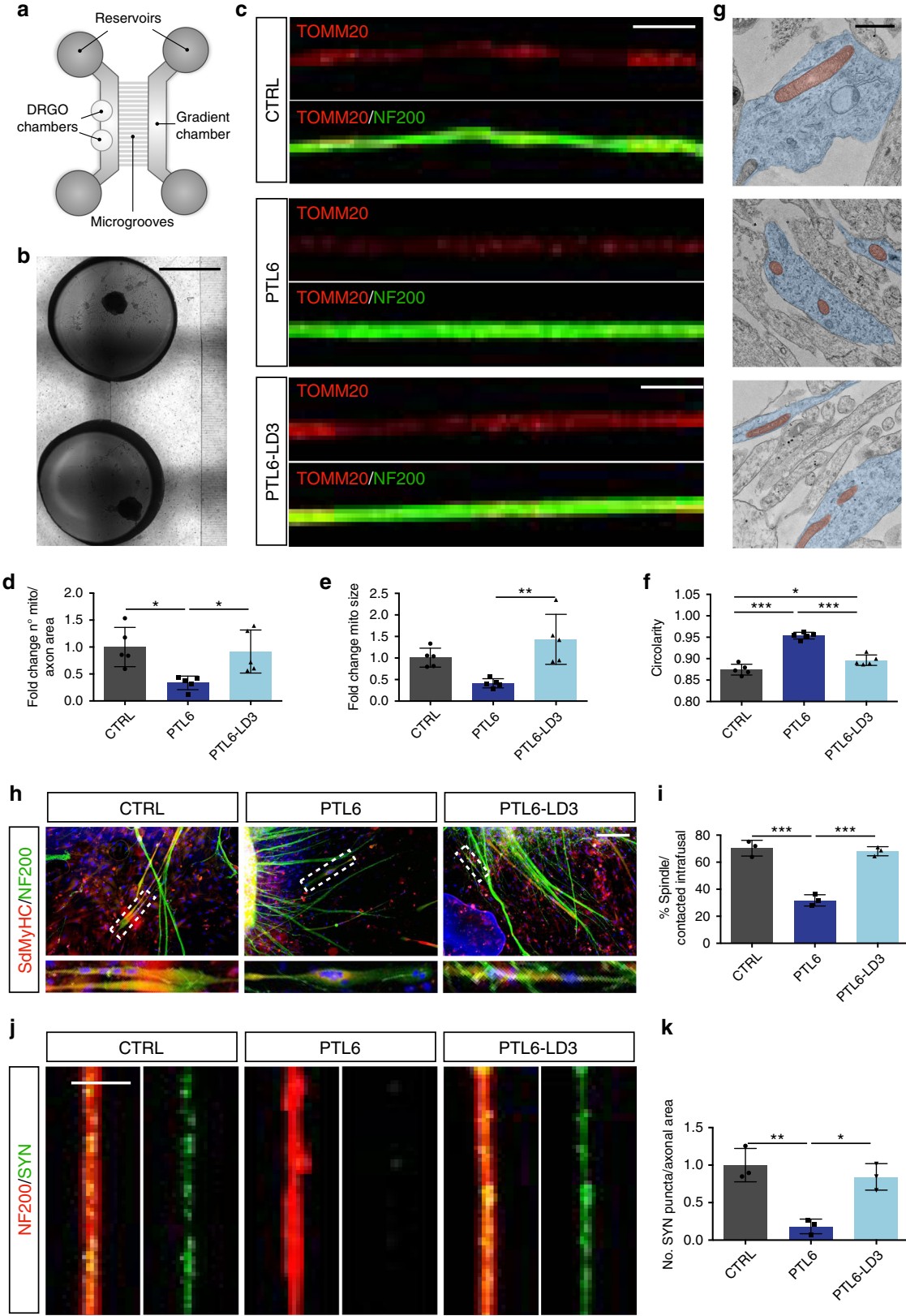

To assess whether these mitochondrial defects might contribute to the impairment specifically of proprioceptive axonal terminals to form muscle spindle with intrafusal muscle fibers, we seeded DIV 16 patient and PTL6-LD3 DRGOs on intrafusal muscle fibers. Four weeks later, patient DRGOs showed a marked reduction in muscle spindle generation as the vast majority of intrafusal fibers failed to form contacts with axonal protrusions, through annulospiral junctions (Fig. 7h, i). In contrast, isogenic PTL6-LD3 DRGOs showed a recovery in the potential to generate muscle spindles by the formation of annulospiral axonal terminals around intrafusal muscle fibers (Fig. 7h, i). Maturation of DIV 50 FRDA patient DRGOs, maintained in co-culture with

**Fig. 7 Axonal mitochondrial behavior and muscle spindle generation in untreated and targeted patient DRGOs. a** Illustration of the microfluidic device with the two lateral channels connected by a set of horizontal microgrooves of length 935 μm, 5 μm width and 6 μm height. The proximal lateral channel includes two open chambers where DRGOs can be contained and cultured. **b** Representative image of DRGOs seeded in the lateral chambers of the microfluidic device. Scale bar, 1 mm. **c** Representative images of DRGO axons within the microgrooves stained for TOMM20 (red) and NF200 (green) to visualize the number and morphology of mitochondria. Scale bar, 5 μm. **d–f** Analysis of number (**d**), size (**e**) and circularity (**f**) of TOMM20 + mitochondria in axons of control, PTL6 and PTL6-LD3 DRGOs. Mean ± s.d., $n = 5$ independent experiments, 4–6 organoids/line/experiment. *$P < 0.05$; **$P < 0.01$; one-way ANOVA with Bonferroni correction. **g** Representative electron microscopy images showing the morphology of axonal mitochondria (red). Scale bar, 1 μm. **h, i** Representative immunofluorescence (**h**) and quantitative analysis (**i**) of muscle spindles in 4 weeks co-cultures between intrafusal muscle fibers and DRGOs from control (CTRL), PTL6 and isogenic PTL6-LD iPSCs. Mean ± s.d., $n = 3$ independent experiments, 4 organoids/line/experiment. ***$P < 0.001$; one-way ANOVA with Bonferroni correction. Scale bar: 100 μm. **j, k** Immunofluorescence for the synaptic vesicle marker Synapsin (green) within axonal NF200 (red) (**j**) and quantification of Synapsin puncta (**k**) on PTL6 and its isogenic line. Mean ± s.d., $n = 3$ independent experiments 3 organoids/line/experiment. *$P < 0.05$; **$P < 0.01$; one-way ANOVA with Bonferroni correction. Scale bar, 5 μm.

intrafusal fibers, was assessed by immunocytochemical analysis. Patients DRGO axons appeared in advanced state of degeneration with marked reduction of Synapsin (SYN) positive puncta, confirming the strong impairment in FRDA axons (Fig. 7j, k). Remarkably, these axonal deficits were extensively rescued in PTL6-LD3 DRGOs (Fig. 7j, k).

Collectively, these results confirmed that patient PTL6 DRGOs axons are populated with fewer and morphologically altered mitochondria. Both number and morphology of mitochondria within axons were restored after deletion of *FXN* intron 1 in PTL6-LD3 DRGOs as was the ability to form muscle spindles and synaptic structures.

## Discussion

Herein we developed a small molecule-based protocol to differentiate human iPSCs into organoids that reconstitute the geometry of somatic DRGs. This 3D protocol generated organoids that contain the principal sensory neural subtypes and glial cells present in somatic DRGs. These findings might suggest a significant role for juxtacrine and/or paracrine signaling, that might include self-reinforcing and cross-repressive activity within the ganglia, and that contributes to neuronal diversification. scRNA-seq analysis showed that DRGOs assembled sensory neurons and their precursors at different stage of differentiation. Moreover, nociceptors and their precursors were the least represented in DRGOs. This is in line with the maturation of sensory neurons in vivo which occurs relatively late during embryonic gestation with the *NTRK1*[+] nociceptors being the latest neurons to be specified[45]. In addition, maturation of sensory neurons strongly relays on extrinsic cues, like NGF and other secreted factors, released by their cellular targets whose axons are in direct connection with[46,57]. In this perspective, final specification of DRGO sensory neurons is a relative long process in vitro and it likely requires a direct connection with their post-synaptic targets. Nonetheless, we showed that a large fraction of DIV80 DRGO sensory neurons clearly present extensive functional features illustrated by their excitability, synaptogenic competence and strong capacity to organize complex structures. These features are exemplified by the annulospiral wrapping of intrafusal fibers to produce a muscle spindle. These findings coupled with the observed presence of punctuated vGLUT1 staining along axon-like fascicles at muscle spindle terminals, as well as that of clathrin-coated vesicles, may indicate the presence of a functional glutamate neurotransmitter machinery which is essential for maintaining the stretch sensitivity of muscle spindle endings. However, ultimate evidence for the functional communication between the neuronal and peripheral elements hinges on the ability to record neuronal response upon physical muscle activity, which might be potentially achieved with microdevices equipped with mechanical stimulation[58]. The reconstitution of this human sensory pathway offers a convenient and powerful setting for studying the cellular processes in proprioceptive sensory neurons in native-like communication with their peripheral post-synaptic elements. This demonstration points at the feasibility of reconstructing a human proprioceptive network in vitro, and should accelerate future endeavors to generate other sensory circuits such as those of mechanosensory end-organs and nociceptive free endings in co-culture with skin cells[59,60]. Together these results render the DRGOs an attractive model for investigating neurological pathologies affecting the DRG in a robust, human in vitro tissue. In this work, we presented and validated stable and reproducible cultures of FRDA patient DRGOs that displayed various aspects of the pathological phenotype and were capable of withstanding several months in culture (>80 DIV). FRDA DRGOs presented impaired survival rates caused by enhanced and widespread cell death. This is echoed by serious defects at morphological and molecular levels in both the cytosolic and mitochondrial compartments as a result of *FXN* silencing. Furthermore, we modified the general design of a microfluidic chip to easily plate and culture DRGOs. This novel design facilitated segregation and constricted development of their axonal projections into an ordered array, which allowed, not only the independent interrogation of the discrete neural compartments, but also a single axon resolution. With this setup we demonstrated that FRDA axons have a reduced number of mitochondria with altered size and morphology.

Here, we confirmed that the targeted excision of the GAA repeats with two different genomic deletions in patient iPSCs, strongly ameliorated the molecular and cellular disease phenotype in DRGOs. Nonetheless, the removal, exclusively of the expanded GAA tract, was less effective in reverting the pathological hallmarks, including *FXN* re-expression, axonal spreading and synaptic machinery organization, with respect to the larger deletion which ablated most of *FXN* intron 1. This disparity in disease recovery between the two genomic modifications mirrors their dissimilar efficacy in restoring *FXN* expression. Indeed, the strategy that was most successful in restoring frataxin transcript and protein levels comparable to those found in healthy conditions was the near-complete removal of the intronic regions flanking the GAA repeats. Since it is well documented that *FXN* intronic regions flanking the GAA repeats are in a transcriptionally non-permissive heterochromatin state, we anticipated that their additional excision would be necessary for full *FXN* reactivation. Epigenetic profiling of the repressive marks H3K9me3, and DNA methylation on *FXN* promoter confirmed this hypothesis, showing that removing exclusively of the expanded GAA tract impacts only partially the abnormally high H3K9me3 accumulation within the *FXN* intron 1 chromatin and DNA methylation on promoter. These results provide direct evidence that the repressed chromatin state of the intronic regions flanking the GAA tract negatively impact *FXN* expression. Although the two genomic deletions were performed in

undifferentiated iPSCs, the repressive state of the chromatin in the *FXN* intronic region remained unchanged during subsequent cell expansion and differentiation in DRGOs. These observations indicate that even after removal of the GAA expanded tract the high levels of H3K9me3 and DNA methylation remain very stable over a long period of time in culture and even after neural cell differentiation.

The elimination of nearly 10 Kb of *FXN* intron 1 might raise some concerns with regards to the transcriptional regulation of the gene since some regulatory regions were mapped within this region[61,62]. While the ablation of these elements, expressed episomally in mouse cells reduce the levels of synthetic reporters, there is evidence that this might not be the case in human cells[63]. In fact, deletion of the E-box motif, close to the GAA tract previously implicated in regulating *FXN* expression in C2C12 cells, did not interfere with *FXN* expression in human blood cells[61]. In addition, endogenous frataxin levels in DRGOs from controls and patients with the long deletion were comparable, suggesting that transcriptional regulation of the gene in DRGOs was not evidently affected. Similarly, the targeted deletion exclusively of the GAA repeats with a short deletion was also able to elicit a significant *FXN* reactivation, mostly evident in the patient with a relatively mild pathological expansion. This finding leaves uncertain the efficacy of the short deletion in reverting the full FRDA symptomatology, and to what extent. Thus, it is plausible that short deletions will not be sufficient to completely revert the severe pathological deficits associated with large expanded GAA tracts. Accordingly, in DRGOs from the patient with the longer GAA tract (PTL6) and higher repressive epigenetic state, the short deletions were less efficient in normalizing the several FRDA deficits. Thus, the targeted elimination of the GAA tract together with its intronic regions should be considered an universal approach for complete *FXN* gene reactivation, even in patients with severely long GAA tracts.

Altogether, we described a 3D protocol to generate iPSC-derived organoids that recapitulate many aspects of human DRGs including transcriptional signature, sensory neuronal subtype diversity, robust functional activity, and the capacity for peripheral target innervation. Furthermore, this system revealed several pathological deficits caused by *FXN* silencing, such as compromised survival and function of DRGO sensory neurons. Thus, this system represents an unprecedented cellular platform for modeling and testing therapeutic strategies related to FRDA and, more generally, for other pathologies afflicting the peripheral nervous system such as chronic pain or peripheral nerve damage. Finally, we showed that DRGO cultures can be coupled with microfluidic technologies to more finely organize and compartmentalize DRGO neuronal networks. Moreover, microdevices can be exploited to spatially organize connections between DRGOs, spinal motor neurons and muscle cells to reconstitute more complex neuronal circuits such as the reflex arc.

## Methods

**Cell cultures and DRGO differentiation**. Healthy control fibroblasts (Neof2) were obtained from ATCC, respectively. FRDA and normal donor patient fibroblasts were obtained from the Franco Taroni's lab at the IRCCS Carlo Besta Neurological Institute and described previously[64,65]. Human iPSCs used in this study where reprogrammed using the CytoTune®-iPS 2.0 Sendai Reprogramming Kit (Invitrogen A16517) and maintained in feeder-free conditions in a mTeSR™1 (Stem Cell Technologies 85850) medium supplemented with 1% Penicillin-Streptomycin (Sigma-Aldrich P0781 Stock 10,000 units penicillin and 10 mg streptomycin/mL). Cells were grown and maintained in 6-well culture plates coated with Matrigel® hESC-Qualified Matrix, LDEV-free (Corning 354277).

To generate iPSC-derived dorsal root ganglion organoids (DRGOs), $9 \times 10^3$ iPSCs, in 150 µL mTeSR with Y27632 100 µM, were seeded into low-adhesion V-bottom 96-multiwell plates (Thermo 277143). The following day, KSR medium was added (DMEM-F12 with 15% KSR, 1% of each: pen/strep; Non-Essential Amino Acids (NEAA Thermo 11140035) with β-mercaptoethanol (100 µM, Thermo

31350010) and L-Glutamate (2 nM). The following day (DIV 0) the KSR medium was supplemented with SB431542 10uM (Merck Sigma S4317) and LDN193189 100 µM (Stemgent 04-0074). The medium was subsequently changed every 48 h. From DIV 4 to DIV 9, in addition to SB and LDN, CHIR99021 3 µM (Miltenyi 130-103-926); SU5402 3 µM (Merck SML0443) and DAPT 10 µM (Merck D5942) were added. During these days KSR medium was gradually switched to N2 medium (Neurobasal medium (Thermo 21103049); N2 (Thermo 17502-048); pen/strep; NEAA, and L-Glutamine 2 mM). On DIV 10 the base medium was changed to 100% N2 medium plus recombinant human Brain Derived Neurotrophic Factor 10 ng/ml (BDNF Peprotech 450-02); Glial-Derived Neurotrophic Factor 10 ng/ml (GDNF Peprotech 450-10); Nerve Growth factor 10 ng/ml (NGF Merck N6009); Neurotrophin-3 10 ng/ml (NT-3 Peprotech 450-03) and Ascorbic Acid 200 µM (AA Merck 49752). On Day 14, organoids were plated into Matrigel®-coated 24-well plates with and without glass coverside, directly onto cultures of muscle spindle fibers or in microfluidic devices and maintained in N2 medium supplemented with the maturation factors. Half of the medium was replaced with fresh medium every 72 h until samples were collected for analysis or for up to 90 days. Two fluorodeoxyuridine (FUDR) treatments were performed from DIV 17 to 26.

Human primary myoblasts (#48046 and #105809) were obtained from Telethon Network of Genetic Biobanks and were maintained and differentiated into intrafusal muscle fibers as previously described (Hippenmeyer et al.[50]). In brief, $2–4 \times 10^4$ myoblasts/cm² are plated on Matrigel-coated glass cover slip in DMEM, 20% FBS (Merck F7524; L-Glutamine 10 µg/mL (Merck G6654); Human Insulin 10 µg/mL (I9278); FGF-b 25 ng/mL (130-104-923); EGF 10 ng/mL (Thermo PHG031). When cells reached 70% confluency, myotube fusion was promoted by removing FBS, FGF-b and EGF. After 72 h, NRG1 1 nM (R&D Systems 396-HB) was added to the differentiation media to promote the generation of muscle spindle fibers.

**Repeat-primed PCR**. The assay is based on the method by Cagnoli and colleagues[66]. In brief, a fluorescent primer was designed in the *FXN* locus-specific region upstream of the unstable GAA peat and a 5′ tail that was used as an anchor for a second reverse primer, preventing progressive shortening of the PCR products along cycles. PCRs were performed with the following general conditions: 200–1000 ng of gGDNA, 800 mmol/L of locus-specific (forward primers: 5′-HEX-gggattggttgccagtgcttaaaagttag) and 40 mmol/L of repeat-specific (reverse: 5′-tacg-catcccagtttgagacgttcttcttcttcttcttcttcttc) primer and 800 mmol/L of the "common" flag primer (5′-tacgcatcccagtttgagacg), 200 µmol/L dNTPs, 1.5 mmol/L MgCl₂, and 1U of TaqGold in 1X specific buffer (Applera Italia, Monza, Italy). PCR parameters were: initial denaturation 7 min at 95°, 40 cycles consisting in 45 s at 95°, 1 min at 60°, 3 min at 72°, and a final extension of 10 min at 72°.

**Deletion of the expanded GAA tracts in FRDA iPSC lines**. sgRNA pairs were designed in regions of intron 1 of the FXN gene (Gene ID: 2395). The program CRISPOR http://crispor.tefor.net/ was used to design three pairs of sgRNAs per category (Supplementary Table 1). Each sgRNA was independently assessed for its activity of INDEL rate using the T7 Endonuclease I assay. With this analysis, we were able to assemble two pairs of sgRNAs with the highest DNA genomic deletion activity (SD-up-gRNA2/SD-down-gRNA1 and LD-up-gRNA2/LD-down-gRNA3). Then, PLT6 and PTS36 FRD iPSC lines were co-transfected with the vectors LV-U6-sgRNA-EF1α-Blast (one for each selected sgRNA) (Rubio et al. 2016)[67] and the pCAG-Cas9-Puro using the Lipofectamine Stem Cells Transfection Reagent (ThermoFisher Scientific)[68]. Co-transfected colonies were then selected by the combination of puromycin (1 µg/ml, Sigma) and blasticidin (10 µg/ml, Thermo-Fisher Scientific) and then isolated through single colony picking. Finally, cell clones with the correct genomic deletions were assessed by genomic PCR analysis followed by Sanger sequencing.

**DNA analysis**. Wizard® SV Gel and PCR Clean-Up System (Promega A9281) was used to purify samples. GoTaq® PCR Core System (Promega M7660) or Phusion® High-Fidelity PCR Kit (BioLabs E553L) were utilized to amplify samples as needed. Primers for PCR analysis are listed in Supplementary Table 2.

**Gene expression analysis**. RNA from 10 to 25 DRGOs was extracted and purified with the QIAGEN® RNeasy® Micro Kit (cat. No. 74004). 0.5–1 µg of RNA was reversed transcribed with ImProm-II™ Reverse Transcription System (Promega A3800). One microliter of the reverse transcribed cDNA was amplified in 25 µl of reaction mixture containing Taq polymerase buffer (Fisher BioReagents), 0.2 mM dNTPs (Finnzymes OY, Espoo, Finland), 1 U Taq polymerase (Fisher BioReagents). The thermal profile consisted of an initial denaturation step for 5 min at 95 °C, followed by 30 cycles of 30 s at 95 °C, 30 s at the specific annealing temperature and 40 s at 72 °C, followed by a final extension of 10 min at 95 °C. The thermocycler C1000™ thermal cycler (BioRad) was used for qPCR measures. Primers for gene expression analysis are listed in Supplementary Table 2.

**Transcriptomic analysis**. RNA libraries were generated starting from 1 µg of total RNA isolated from DRGOs, the quality of which was assessed by using a Tape Station instrument (Agilent). To avoid over-representation of 3′ ends, only

high-quality RNA with an RNA Integrity Number (RIN) ≥ 9 was used. RNA was processed according to the TruSeq Stranded mRNA Library Prep Kit protocol. The libraries were sequenced on an Illumina HiSeq 3000 with 76 bp stranded reads using Illumina TruSeq technology. Image processing and base calling were performed using the Illumina Real Time Analysis Software.

Fastq files were mapped to the hg38 mouse reference genome with the Bowtie2 aligner. Differential gene expression and Functional enrichment analyses were performed with DESeq2[69] and GSEA[70], respectively. Statistical and downstream bioinformatics analyses were performed within the R environment. Gene expression heatmaps were produced with GENE-E (Broad Institute). Data were deposited in the NCBI Gene Expression Omnibus repository with the GSE133755 GEO ID.

RNA-Seq human data mining was performed using data from NCBI GEO/SRA repositories: spinal motor neurons (GEO GSE93939)[71]; prefrontal cortex (GEO GSE102556)[72]; cerebellum (GEO GSE97942)[73]; DRG (SRA SRP077657), iPSCs (GEO GSE120081); iPSC-derived sensory neurons (GEO GSE26867)[35]; fibroblast-derived nociceptors (SRA SRP165240)[38].

GO aggregated categories were produced by combining multiple GO datasets: *sensory neuron development* (GO Sensory Organ Development, GO Sensory Organ Morphogenesis), *Sensory Perception* (GO Sensory Perception Of Taste, GO Response To Mechanical Stimulus, GO Sensory Perception Of Pain, GO Response To Pain, GO Detection Of Mechanical Stimulus Involved In Sensory Perception, GO Sensory Perception Of Temperature Stimulus, GO Chemosensory Behavior, GO Detection Of Chemical Stimulus Involved In Sensory Perception Of Taste, GO Detection Of Light Stimulus Involved In Sensory Perception, GO Detection Of Mechanical Stimulus Involved In Sensory Perception Of Sound, GO Mechanosensory Behavior, GO Detection Of Temperature Stimulus Involved In Sensory Perception, GO Regulation Of Sensory Perception, GO Sensory Perception, GO Sensory Perception Of Chemical Stimulus, GO Sensory Perception Of Light Stimulus, GO Sensory Perception Of Mechanical Stimulus), *Schwann cells/astrocytes* (GO Compact Myelin, GO Myelin Assembly, GO Myelin Maintenance, GO Myelin Sheath, GO Positive Regulation Of Myelination, GO Regulation Of Myelination, GO Sphingomyelin Metabolic Process), *peripheral sensory neuron genes* (GO Sensory Perception Of Taste, GO Response To Mechanical Stimulus, GO Sensory Perception Of Pain, GO Response To Pain, GO Detection Of Mechanical Stimulus Involved In Sensory Perception, GO Sensory Perception Of Temperature Stimulus, GO Chemosensory Behavior, GO Detection Of Chemical Stimulus Involved In Sensory Perception Of Taste, GO Detection Of Light Stimulus Involved In Sensory Perception, GO Detection Of Mechanical Stimulus Involved In Sensory Perception Of Sound, GO Mechanosensory Behavior, GO Detection Of Temperature Stimulus Involved In Sensory Perception, GO Regulation Of Sensory Perception, GO Sensory Perception, GO Sensory Perception Of Chemical Stimulus, GO Sensory Perception Of Light Stimulus, GO Sensory Perception Of Mechanical Stimulus, GO Sensory Organ Development, GO Sensory Organ Morphogenesis).

**Single-cell RNA sequencing**. DRGOs were treated with Trypsine-EDTA 0.25% (Gibco) and DNaseI 200 μg/ml (Sigma) for 30 min at 37 °C. Then they were gentle triturated pipetting up and down, washed with PBS and counted with Trypan Blue (Invitrogen). Single-cell RNA sequencing was performed through the Chromium platform (10X Genomics). Cells were separated into droplet emulsion using the Chromium Single Cell 3′ Solution (V3.0) and Single-cell RNA-seq libraries were prepared according to the Single Cell 3′ Reagent Kits User Guide (V3.0). Libraries were sequenced on a Novaseq 6000 flowcell (Illumina), with a depth of 50 K reads/cell. FASTQ reads were aligned to the GRCh38 human reference genome with cellranger count (10x Genomics) with default parameters, and human Gencode v33 annotations (PMID:30357393). Gene count matrix log-normalization (10,000 scale factor), gene clustering, dimension reduction analysis (UMAP), differential gene expression analysis, and plotting were performed with Seurat (PMID:31178118) within the R environment (R Core Team (2020). R: A language and environment for statistical computing. R Foundation for Statistical Computing, Vienna, Austria. URL: https://www.R-project.org/). Functional enrichment of GO biological processes datasets was performed with DAVID (PMID:17784955).

**CHIP-qPCR assays**. iPSC-derived DRGOs were fixed incubation in 1% formaldehyde for 15 min at room temperature. Crosslinking was then quenched by addition of glycine to a final concentration of 125 mM Glycin. Subsequently, cells were washed in PBS, harvested by scraping, and resuspended in SDS lysis buffer (1% SDS, 10 mM EDTA, 50 mM Tris), containing protease inhibitors (Roche). The chromatin was sheared by sonication using a Bioruptor (Diagenode) sonicator for 10 min in 30 s ON/OFF cycles. Samples were centrifuged at 16,200 g to remove debris and the DNA-concentration was determined by a Nanodrop Spectrophotometer. Immunoprecipitations with rabbit anti-H3K9me3 (abcam-cod.8898), H3K9ac (Diagenode, cod. C15410004), rabbit anti-H3 (Abcam-cod.1792) control GFP antibodies (Thermo Fisher, A6455) were done using 30 μg of chromatin and 5 μg of antibody per assay as previously described[74]. DNA sequences were quantified by real-time PCR (primers are listed in Supplementary Table 3). Quantities of immunoprecipitated DNA were calculated by comparison with a standard curve generated by serial dilutions of input DNA. The data were plotted as means of at least two independent ChIP assays (biological replicates) and three independent amplifications (technical replicates).

**meDIP-qPCRs**. For meDIP 1 μg of purified gDNA, using qiAMP DNA mini kit (Qiagen, cat. 51304) was used for control, patients and isogenic corrected lines.

In brief, for methylated DNA immune precipitation and purification using MagMeDIPqPCR kit (Diagenode, cod. C02010020). First, gDNA was sonicated to obtain a fragment size between 250 and 400 bp, then, it was denaturated to ssDNA and immunoprecipitated using the antibody provided by the kit. The next day, immunoprecipitated DNA and Input were purified and eluted. For quantitative analysis by real-time PCR DNA was diluted 1:10 (primers are listed in Supplementary Table 3).

**Microfluidic fabrication**. The microfluidic chambers were customized by MiMic Lab (PoliMI, Italy). Microfluidic devices were fabricated through soft-lithography of PDMS on master molds. The design as shown in Fig. 7a features two lateral channels (1.5 mm width, 50 μm height, 7 mm length), one of them with two 2 mm chambers to house the DRGOs, connected by an array of ~130 microchannels for axon guidance (5 μm width, 6 μm height, 625 μm length). The design of the channels was printed on a high-resolution photomask (64,000 DPI) for subsequent master mold fabrication steps. Master molds were fabricated through standard two-layer photolithography of SU-8 (Microchem, Germany) on silicon wafers. In particular, the first layer of SU8-2005 with a thickness of 6 μm was exposed on a mask-less laser writer (MLA100, Heidelberg, Germany) to obtain the microchannel design alone. Subsequently, the second layer of SU-8 2050 was deposited to a thickness of 50 μm and exposed through a standard mask aligner and photomask to obtain the channels. To obtain microfluidic layers, PDMS was poured on silicon wafers at a pre-polymer to curing agent mixing ratio of 10:1 (w/w) and cured at 65 °C for 3 h. Cast PDMS was peeled-off the molds, trimmed and through-holes were punched to obtain culture chamber access ports (8 mm diameter) and synaptic chamber access ports (3 mm diameter). Finally, PDMS microfluidic layers were plasma bonded (Harrick Plasma, USA) to glass cover slides and stored until use.

**Immunocytochemical assays**. For immunocytochemical analysis, DRGOs were fixed for 20 min at room temperature in 4% paraformaldehyde in PBS, permeabilized for 30 min in PBS containing 0.1% Triton X-100 and 10% normal goat serum and incubated overnight at 4 °C in PBS containing 10% normal goat serum and primary antibodies. Then cells were washed three times with PBS and incubated for 2 h at room temperature with secondary antibodies. For DRGO cryosection, organoids were fixed in 4% paraformaldehyde for 40 min at room temperature and washed with PBS three times at room temperature before allowing to deposit in 30% sucrose at 4 °C. The tissues were embedded and sectioned and stained as described in ref. [40]. A list of antibodies used can be found in Supplementary Table 4. The percentages of the cells that were positive to specific markers were determined from at least three independent experiments for each condition. Images for quantification were selected randomly. First, the number of nuclei was counted, followed by counting the number of cells expressing the markers of interest in at least five fields per condition. To quantify the immunopositive area (expressed in pixels) pictures in each coverslip/sample were taken and the area of interest was calculated using the ImageJ software and normalized on nuclear counts.

**Electron microscopy**. Electron microscopy analysis on intrafusal muscle fibers–DRGO co-cultures were performed as described previously[42]. In brief, cells were fixed in 0.1 M Sořensen phosphate buffer (pH 7.2), containing 2.5% glutaraldehyde and 0.5% sucrose, for 4–6 h. The specimens were then washed in a solution containing 1.5% sucrose in 0.1 M Sörensen phosphate buffer for 6–12 h, post-fixed in 2% osmium tetroxide, dehydrated and embedded in Glauerts' embedding mixture. Semi-thin sections were cut on an Ultracut UCT ultra-microtome (Leica, Wetzlar, Germany), and stained with toluidine blue. From the same tissue blocks, ultra-thin sections (50–70 nm) were also cut using the same ultramicrotome and placed on copper grids. Grids were then stained with uranyl acetate and lead citrate and observed on a JEM-1010 transmission electron microscope (JEOL, Tokyo, Japan) operating at 80 kV and equipped with a Mega-View-III digital camera and a Soft- Imaging-System (SIS, Muñster, Germany) for the computerized acquisition of the images.

**Western blotting**. Protein extract from 20-36 DRGOs were prepared in RIPA buffer (10 mM Tris-HCl pH 7.4, 150 mM NaCl, 1 mM EGTA, 0.5% Triton and complete 1% protease inhibitor mixture, Roche Diagnostics). Protein was separated on SDS-polyacrylamide gel, transferred to a nitrocellulose membrane and probed with a primary antibody followed by horseradish-peroxidase-conjugated secondary antibody (1:5000, Dako). ACTIN was used as normalizer. A list of antibodies used can be found in Supplementary Table 4. Finally, bands were visualized using ECL chemiluminescence (GE Healthcare)

**Patch-clamp electrophysiological recordings and single-cell-RT-PCR**. Organoids cultured on glass coverslips were placed in the registration chamber which

was, then, mounted on the stage of the BX51WI Olympus Upright microscope. Samples were continuously perfused with artificial cerebrospinal fluid (nACSF) (125 mM NaCl; 2.5 mM KCl; 1.25 mM NaH$_2$PO$_4$; 2 mM CaCl$_2$; 25 mM NaHCO$_3$; 1 mM MgCl$_2$; and 10 mM D-glucose; bubbled with 95% O$_2$; 5% CO$_2$ pH 7.3) with a flowrate of 2–3 mL/min. Electric currents were, then, registered via whole-cell patch clamp utilizing glass pipets electrodes filled with an electrolytic solution composed of 10 mM NaCl; 125 mM KH$_2$PO$_4$; 10 mM HEPES; 0.5 mM EGTA; 2 mM MgCl$_2$; 55 mM Na2-ATP; 0.5 mM Na-GTP (pH 7.3). Cell resting potential was determined in whole-cell configuration 2 min after patch rupture in current-clamp mode. Firing profile was recorded in current-clamp configuration applying progressive depolarizing steps of 10 pA. For spontaneous EPSCs recording, cells were recorded in whole-cell voltage-clamp configuration in gap-free mode. Currents where low-pass-filtered at 2 kHz and acquired online at 5–10 kHz with pClamp10 hardware and software (Molecular Devices). No correction of liquid-junction potential was applied. After the patch clamp a negative pressure was applied to the patch pipette to slowly withdraw the cell from the rest of the culture. The glass tip was broken in the PCR collection tube containing 7 μl water. cDNA was obtained by the REPLI-g WTA Single Cell Kit (Qiagen) following manufacturer's instructions, and neuronal subtypes identification was performed by PCR for SPP1, NTRK1, NTRK2, KNTR3, CACNA1H, PLXNC1, PVALB (primers are listed in Supplementary Table 3). Data were analyzed with Clampfit10 and GraphPad Prysm7.

**ROS FACS analysis**. Cells were treated for 15 min with H2DCFDA (DCF, Invitrogen) 1:5000 diluted in PBS. Data were acquired using a FACSCantoII (BD Bioscience) and were analyzed using FCS Express 7 (De Novo Software). Unstained samples were employed to set gates.

**Statistics and reproducibility**. All data are represented as the mean calculated between different cultures and the variation between cultures is depicted as the standard deviation (SD). For each experiment, "n" indicates the number of independent experiments, when non indicated it supposed to be "$n = 3$ independent experiments". Analyses of significant differences between means were performed using two-tailed Student's t-tests. Statistical significance was set at: *$p < 0.05$. Comparison of parameters values related to different culture types were performed by means of a statistical test for multiple independent variables (Kruskal–Wallis test + Mann–Whitney as post hoc assessment).

**Reporting summary**. Further information on experimental design is available in the Nature Research Reporting Summary linked to this paper.

## Data availability
Data generated during the study are available in the NCBI GEO public repository with accession no. GSE133755 and GSE148212. Source data are provided with this paper.

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

## Acknowledgements

We are thankful to D. Bonanomi, L. Muzio, A. Menegon and C. Taveggia for sharing of reagents, all members of the Broccoli's lab for helpful discussion. Device manufacturing was partially performed at PoliFAB—the micro- and nanofabrication facility of Politecnico di Milano. The bank of "Cells, tissues and DNA from patients with neuromuscular diseases", member of the Telethon Network of Genetic Biobanks (project no. GTB18001), funded by Telethon Italy, provided us with specimens. This work was supported by the European Research Council (AdERC #340527, PoC #842423) and Friedreich's Ataxia Research Alliance (FARA).

## Author contributions

P.G.M., conceptualization, methodology, supervision, writing; S.M., conceptualization, investigation, methodology; M.L., investigation, writing, L.M., data curation; M.Z., investigation; P.T.T.V., investigation; S.B., investigation; M.J.M., investigation; C.P., investigation; G.C., investigation; A.I., investigation; M.P., investigation; C.C., investigation; S.G.G., investigation; P.P., investigation; C.G., investigation; F.T., data curation; F.N., data curation; M.R., conceptualization, supervision, data curation; V.B., conceptualization, supervision, writing, funding acquisition.

## Competing interests

The authors declare no competing interests.
