## [Peer Review File · Nature Communications]

Reviewers' comments:

Reviewer #1 (Remarks to the Author):

In this manuscript by Mazzara et al., the authors developed a new system to model the sensory neurons affected in FRDA. In this work, the authors used patient derived iPSCs to generate dorsal root ganglia organoids (DRGOs). The first part of the manuscript is dedicated to characterizing the DRGOs. The patient derived DRGOs recapitulate neuronal impairment in survival, axonal pathogenic morphology and mitochondria dysregulation. The authors also established connectivity between DRGO sensory neurons and muscle fibers to recapitulate the proprioception sensory pathway in vitro. In the second part of the work, the authors report the excision of FXN intronic region by CRISPR/Cas9 in two different ways; 1- excision of the GAA tract only (PTL6-SD10) or 2- removal of the FXN intron-1 (PTL6-LD3). The authors then described the rescue of the previously identified phenotypes, which were only partial in the short deletion. An elegant approach used in this work is the addition of the microfluidic chamber device to evaluate mitochondrial health. The manuscript is well written and organized and the data is presented in a clear fashion. The overall story is intriguing and novel, and the subject is very relevant to the field. However, there are concerns that should be addressed before this manuscript can be accepted for publication.

Major Issues:

1. The authors used CRISPR/Cas9 targeted for deleting the GAA repeats in two different ways: one removing only the repeats (Short Deletion - SD), and a second removing FXN intron 1 (Long Deletion - LD). Even though the use of CRISPR/Cas9 is a pertinent approach, the restoration of the GAA repeat size to control levels in the "short deletion" iPSCs would be a more adequate control for this study. It should be used in the mitochondria, axon and cell death assays to compare the rescue of the phenotypes to the other 2 deletions.

2. A main accomplishment of this work in the development of dorsal root ganglia organoids, however, the composition of the organoids was not completely described. Based on the results presented in Figure 2D, about 70% of total cells are neurons (BIII-Tub), which raises the question of what would the remaining 30% be. Single-cell PCR for not only neuronal, but also glial markers should help elucidate this issue.

3. Finally, as described in this study "the removal of exclusively the expanded GAA tract was less effective in reverting the pathological hallmarks with respect to the larger deletion which ablated most of FXN intron-1". The authors hypothesized that the inability to restore FXN expression in the removal of expanded GAA tract cells was due to epigenetic mechanisms. There are 2 issues that should be addressed in the study of this hypothesis:

a. The authors did not explore the possibility of the shorter deletion impacting the splicing of the intronic region. To exclude this possibility, a qRT-PCR should be performed to show that deletion of repeats alone does not disrupt the splicing of FXN. This should be carried out for all alleles (expanded, control, LD and SD). If negative, this should strengthen the author's claim regarding the mechanism of FXN repression in the shorter deletion.

b. It was not clear why the authors chose to investigate H3K9me3 specifically. Therefore, there is a need for an epigenetic screening as H3K9me3 may not be the only repressive factor involved in the FDRA pathogenesis.

Minor Issues:

1. Reference missing from sentence: "The length of the GAA expansion correlates proportionally with the severity of the disease and inversely with the age of onset which can occur from infancy to after 60 years of age."

2. The authors described electrophysiological measurements as well as synapsin and vGLUT analysis for characterization of the DRGOs, which raises the question: were there differences between affected and unaffected organoids? Especially, due to the axonal defect reported in figure 6C and 6D.

3. In the title: "FRDA deficits in Fe-S cluster protein biogenesis and antioxidant response are restored in PTL6-DL DRGOs", please correct PTL6-DL to PTL6-LD

4. In the legend of figure 5 E, please introduce the acronyms UC5', UpSD, DwSD, UC3'.

Reviewer #2 (Remarks to the Author):

In their manuscript, Mazzara and colleagues report: 1) the generation of human induced pluripotent stem cells (iPSCs) from two patients with Friedreich's ataxia (FRDA) and GAA repeat expansions of different lengths; 2) the generation of isogenic controls via the CRISPR-mediated excision of the GAA repeats or of most of FXN intron 1; the differentiation of these iPSCs into what they call dorsal root ganglia organoids (DRGOs), including the establishment of connections with co-cultured muscle spindles; excess cell death, loss of Fe-S proteins and abnormal antioxidant responses in FRDA-derived DRGOs.

This is an interesting study with potential to provide relevant insight into FRDA pathogenesis. There are however, some concerns:

1. No details are provided about the control iPSC line utilized.
2. The morphological differences between DRGOs and the clusters formed in traditional differentiation protocols for SN should be better specified. In both cases cellular aggregates extending long neurites form.
3. In FRDA, proprioceptive neurons are most vulnerable, showing developmental defects, while other SN neurons seem to be affected only later (e.g. Creigh P et al. *Ann Clin Transl Neurol* 2019; 6:1718-1727), so it is important to understand the relative representation of different types of SN in the DRGOs. This is given only approximately. In particular, data suggest that a significant number of neurons are TrkB/TrkC double positives, i.e. their fate as proprioceptors or mechanoreceptors is not yet determined. This requires co-localization experiments of specific markers defining such subpopulation. IF images of better quality should be provided at different magnifications. Single-cell RNAseq may also be used. This is important also because most FXN in DRGs comes from proprioceptors (see Piguet F et al. *Mol Ther.* 2018; 26:1940-1952).
4. The excess cell loss and abnormal morphology of FRDA DRGOs is an interesting observation, but it should be better specified. Is there a differentiation deficit? Does it specifically affect proprioceptors? If normally differentiated cells die prematurely, what are their identities, is there a specific vulnerability?
5. Electrophysiology data confirm the neuronal phenotype of most analyzed cells, but again it is unclear what type of neurons they represent. Are there any corresponding, morphology data, i.e. after biocytin injection of clamped cells?
6. In Figure 4 it is not specified if electrophysiology data are from FRDA, controls or isogenic control cells. These should be analyzed separately.
7. The LD vs. SD observation is intriguing, but the only possible explanation is the persistence of inhibitory epigenetic marks in intron 1 after removal of GAA expansions. These may be histone PTMs or DNA methylation, which may last longer and has not been explored. There is some evidence that long repeats may be associated with more extensive DNA methylation in intron 1. The effect of continued culturing of iPSCs allowing more cell divisions should also be investigated.
8. There are several imprecisions about the disease, including the statement that proprioceptive neurons and Purkinje cells are most vulnerable (page 6). This is incorrect, as the most vulnerable cells in the cerebellum are large projection neurons in the deep nuclei, while Purkinje cells are

mostly preserved in number and morphology, except for their axonal endings in the deep nuclei undergoing grumose degeneration, possibly due to loss of target cells.

9. The paper needs a revision for grammar and style, preferably by a native English speaker.

Reviewer #3 (Remarks to the Author):

Extremely interesting manuscript in which the authors developed an original and novel approach to generate organoid cultures of DRG sensory neurons from iPSC. This is the first example of a protocol for DRG organoids in the literature. Furthermore, the authors developed a very elegant experimental design to allow maturation of the DRGO in the presence of myogenic culture, and demonstrates the presence of muscle spindle. Overall, these novel experimental approaches could be important to develop human-derived models for sensory neuropathies. In the current manuscript, the approach is applied to iPSC cells derived from Friedreich's Ataxia (FRDA) patients, a disease that primarily affects the proprioceptive neurons of the DRG. There is a current lack for good FRDA neuronal models, and the iPSC derived DRG organoids appears as a very powerful solution as it combines the complex genetic context of FRDA (large GAA repeat expansions) and the primarily affected cell type.

Overall, this work represents an important advance in the field of stem cell research through the elaboration of a new DRG organoid differentiation protocol. There are however several weaknesses that temper the enthusiasm.

Major comments:

- Although not completely characterized, it is thought that there are multiple regulatory elements in intron 1 of the frataxin gene. Therefore, removing most of intron 1, as proposed in the manuscript, might have deleterious effect on the fine regulation of frataxin expression. In the current manuscript, the authors suggest that full frataxin recovery can only be achieved with the removal of a large portion of the intron 1 (figure 1F-G, text p6). However, other laboratories in several publications have demonstrated that removing the GAA expansion repeat restores the frataxin levels and prevents the heterochromatin epigenetic profile (Tremblay, Napierala, Gottesfeld laboratories). Furthermore, Lai et al. (2018) generated isogenic iPSC lines that differ only in the length of the GAA repeat and demonstrated the restoration of frataxin expression. Could the authors elaborate on these differences and find a plausible explanation? The characterization of multiple clones for each deletion and each iPSC line would probably clarify this point.

- The authors state that "PTL6 iPSCs showed the strongest frataxin reduction both at transcript and protein levels compared to PTS36". However, the western blots and their quantification on Figure 1F-G are in disagreement with this statement. Indeed, frataxin levels appears to be down by approximately 60% in both lines, and if there is a difference, PTS36 appears to have less than PTL6. At the RNA level, there is indeed a clear difference. The authors should address this discrepancy. Again, analyzing several clones might help in clarifying. At the RNA level, PTS36 appears to have 50% frataxin, levels equivalent to heterozygote carriers. At the protein level, both patients have approximately 40% frataxin, inconsistent with the 5-20% reported in the literature. Can the authors elaborate on this aspect?

- The authors convincingly generated DRG organoids that are composed of all neuronal cell types. A technical question: what is the effect of the anti-mitotic treatment on glial cells, considering that these are extremely important for signaling and for myelination?

The immunofluorescence images from 2B and 2C show very convincingly that the cultures are well organized and contains large sensory neurons that are positive for NF200. Considering that the RT-PCR is not quantitative, it is difficult to evaluate to which extent the culture is enriched for any of the 3 neuronal subtypes. The author should perform quantitative RT-PCR or Western Blot experiments to evaluate the enrichment. This statement holds for all the RT-PCR presented in the manuscript, quantitative would be better (Fig. 2, S2, S3). Furthermore, a better choice of specific antibodies or RT-PCR to determine the population of sensory neurons present in their culture would strengthen the manuscript. Indeed, the authors chose Runx3 and TrkC (NTRK3). Runx3 is

specific of proprioceptive neurons during development and its expression stops when the neurons are mature (Marmigère and Ernfors, 2007). TrkC is expressed both by mechanoreceptive and proprioceptive neurons, as reported in Usoskin et al., 2015. Usoskin et al. present new proprioceptive specific genes such as CNTNAP2 and SPP1 that should be used.

- While there is no doubt that the DRGO contain all subtypes of DRG sensory neurons, the authors should select images with a better staining pattern for the different markers (Figure 2G). Based on the images, it is difficult to understand how the authors manage to quantify properly the subtypes, considering the large amount of overlap in the staining. Furthermore, it is difficult to understand how the total percentage can be above 100% (Figure 2H – 114%), except if several neurons express two different receptors (as expected for mechanoreceptive neurons). The percentages given in the text (p. 7-8) do not correspond to the percentage shown on the graph Figure 2H. Indeed, the text states that the highest percentage is 44.5% corresponding to NTRK2, while Figure 2H shows that NTRK3 has the highest percentage. Is there an inversion? 44.5% of neurons are positive for TrkC which is possible since it stains both proprioceptive and mechanoreceptive neurons. It is however incorrect to state that 44.5% are proprioceptive. The percentage of proprioceptive would be the TrkC positive part (44.5%) minus the TrkB positive part (exclusively mechanoreceptive) (35.8%), which would correspond to around 8-9%. Parvalbumin is specific for proprioceptive neurons in the DRG, and Figure 3E-F shows around 15% of Parvalbumin positive neurons, a percentage that is closer to the physiological number. Overall, the transcriptional profile demonstrates nicely that the DRGO present an expression profile similar to DRG albeit probably closest to developmental stages and immature.

- The results obtained in figure 4 are clearly a highlight of the paper, showing the capacity to generate mature DRG organoids capable of muscle spindles forming and that are electrically active. These extremely impressive and innovative results could be strengthened by complementary experiments. Indeed, it would be interesting to identify the type of neurons that forms the muscle spindle, since in the physiological context, both motor and sensory neurons are found. This could be done using a Parvalbumin antibody to demonstrate the presence of proprioceptive neurons. With the current protocol, are muscle spindle sparse or are they found in a reasonable number? Although probably beyond the current manuscript, have the authors tried to myelinate the neurons to achieve better maturation and functionality?

- Figures 5A to 5D show conflicting results with the mRNA and protein expression quantifications inversely correlated. The long repeat line (PTL6) shows lower protein level than the short repeat line (PTS36), as expected (although opposite to the result on the iPSC lines). Unexpectedly, the mRNA quantifications show the opposite. The authors should explain these conflicting results. The protein levels in the PTL6 line are this time in agreement with the levels described in the literature for FRDA patients. The effect of the SD or LD excision on the expression levels of frataxin appears not to be the same as for the iPSC lines, in particular for PTS36-SD23. Furthermore, high variability is seen – could this be a reflection of variation in the differentiation protocol with variable percentage of neurons from one organoid to another? The authors should discuss this issue in the manuscript.

- In Figure 5F-G, the H3K9me3 pattern of PTL6 is different between the two figures. For example, in 5G, at the UC5' site, the level of the mark is higher than any other site, although it is among the lowest in 5F. Could the authors comment these results?

- It appears that the authors accidentally present the exact same graphs in 6B as this is not in agreement with the rest (i.e. should the first graph be PTS36 and derivatives?). It would be worth for the authors to check. Overall Figure 6 shows a nice rescue of the phenotypes after the GAA repeats excision. However, there is a lack in tissue specificity, as the result show that every cell type is affected by FXN depletion, although proprioceptive neurons should be more sensitive than the rest (as explained earlier by the authors). Could the authors comment on this? Clearly the phenotype observed in the dish appears to agree with a developmental component of the disease as proposed more recently in the field. However, the authors also see an increase in death in FRDA DRGO compared to controls. In the text p 12, it is not always clear to which control the authors are referring to. This section should be re-written.

- The authors explore the phenotype of the FRDA DRGO, with a particular focus on mitochondria, as it is known that mitochondria are particularly susceptible to FXN depletion. The authors have

well adapted a powerful technique long known to arrange neuronal axons in microgrooves, allowing to follow organelles movement and follow the direction of the movement, using microfluidic chips. The novelty here is that the authors have designed additional chambers compared to classic neuron adapted microfluidic chips, that permits the cultures of the DRGO. With this system, the authors were able to measure several parameters of mitochondrial morphology and dynamics. The authors demonstrated a decreased mitochondrial number (confirmed by mtDNA/gDNA quantifications in supplementary data), and overall decreased mitochondrial size. The graph and statistics for the circularity should be checked because there does not appear to be any difference between the 3 conditions.

- In the last part of the manuscript, the authors investigated FRDA-related phenotypes such as iron-sulfur clusters (ISC) deficiency and oxidative stress. This is by far the weaker part of the manuscript and would benefit from being reinforced by a few complementary experiments. It is important to demonstrate whether there is an ISC cluster deficiency has this is the primary role of frataxin. The authors demonstrate a very strong decreased level of aconitase (around 90% in the PTL6 mutant), which seems unlikely as this protein is known to be the less sensitive to degradation when deprived of its ISC. There are numerous other proteins that can be tested by western blot to demonstrate indirectly a ISC deficiency, such as Complex I and II of the respiratory chain or Lipoic Acid bound to proteins. The activity of the enzyme (such as SDH or aconitase which are relatively standard measurements) would constitutes a more direct way to evaluate ISC loss. Increase in SOD1 and SOD2 mRNA are interesting results, but are not enough to conclude that there is the presence of an increase oxidative stress in the model. Staining with fluorescent probes specific to oxidative stress such as DHR123 or MitoSOX should be perform. Furthermore, using such probe could allow to investigate the phenotype specifically in the different subtypes of neurones. The tissue specificity is an important question in the FRDA field. Furthermore, the authors could also use a fluorescent probe that allows to investigate mitochondrial health status and activity such as TMRM, TMRE or JC-1. Additional information regarding iron metabolism would be much appreciated as it is part of the classical hallmarks of FRDA.

Minor comments:

1. The bibliography in the introduction is very sparse and an effort should be made to cite primary bibliography.
2. The introduction is not completely updated with the latest development in the FRDA field that are important in the context of the current manuscript
3. Cerebellar Purkinje cell are not extremely affected in FRDA, it is rather the dentate nucleus of the cerebellum that is affected. This should be corrected (p. 6)
4. A throughout reading of the manuscript should be done to remove the multitude of typos or small annotation mistakes that can be found (some examples: legend figure 1B, "FRDA-DS and FRDA-DL" instead of FRDA-SD and FRDA-LD; legend figure 1, "organoids" instead of IPSC; Figure S1E, the oligos used are probably LF1+LR+SR and not LF2 as noted;)
5. Figure 3B – it is extremely difficult to discriminate between the different blues and green, and therefore it is difficult to discriminate between DRGO and IPSC. The colors should be changes
6. The manuscript presents data on 6 different IPCS lines. For the graphs, it would be important to have a code which could differentiate between the different lines. This would considerably help. For example, PTL6 is grey in 5F and then red in 5G. Maybe using hatched bars for one of the lines would help, and color code could then stay.

March 30th, 2020

Manuscript #NCOMMS-19-29226R1 entitled "CRISPR/Cas9 targeted deletions rescue Friedreich's ataxia pathological deficits in a stem cell-based sensory neural circuitry with dorsal root ganglia organoids" by Mazzara, Muggeo et al.

Reviewers' comments:

Reviewer #1 (Remarks to the Author):

In this manuscript by Mazzara et al., the authors developed a new system to model the sensory neurons affected in FRDA. In this work, the authors used patient derived iPSCs to generate dorsal root ganglia organoids (DRGOs). The first part of the manuscript is dedicated to characterizing the DRGOs. The patient derived DRGOs recapitulate neuronal impairment in survival, axonal pathogenic morphology and mitochondria dysregulation. The authors also established connectivity between DRGO sensory neurons and muscle fibers to recapitulate the proprioception sensory pathway in vitro. In the second part of the work, the authors report the excision of FXN intronic region by CRISPR/Cas9 in two different ways; 1- excision of the GAA tract only (PTL6-SD10) or 2- removal of the FXN intron-1 (PTL6-LD3). The authors then described the rescue of the previously identified phenotypes, which were only partial in the short deletion. An elegant approach used in this work is the addition of the microfluidic chamber device to evaluate mitochondrial health. The manuscript is well written and organized and the data is presented in a clear fashion. The overall story is intriguing and novel, and the subject is very relevant to the field. However, there are concerns that should be addressed before this manuscript can be accepted for publication.

Major Issues:

1. The authors used CRISPR/Cas9 targeted for deleting the GAA repeats in two different ways: one removing only the repeats (Short Deletion - SD), and a second removing FXN intron 1 (Long Deletion – LD). Even though the use of CRISPR/Cas9 is a pertinent approach, the restoration of the GAA repeat size to control levels in the "short deletion" iPSCs would be a more adequate control for this study. It should be used in the mitochondria, axon and cell death assays to compare the rescue of the phenotypes to the other 2 deletions.

Answer:

We agree with the reviewer that a further additional control would be an isogenic line with a physiological GAA repeat size. However, we already generated and characterized for this work 4 isogenic cell lines with two different targeted deletions in the two patient iPSC lines. This panel of isogenic cell lines enabled us to obtain a wealth set of data between directly comparable cell lines. One of our goals was to define the extent of rescue using iPSCs with targeted deletions in order to assess the therapeutic potential of this approach. In fact, currently one of the most plausible and feasible therapeutic approaches would be to delete the entire trinucleotide expansion by CRISPR/Cas9 technology, creating a gene locus lacking any GAA repeat. In this

context, our study describes the effects and the extension of the rescue obtainable with this genomic manipulation. Alternatively, any therapeutic approach aiming at restoring the normal GAA repeat size in post-mitotic neurons would be extremely complicated since homology-direct repair (HDR) is not functional in post-mitotic cells. On this light, the iPSC line with a physiological GAA repeat would have been seen as incremental for this type of study.

2. A main accomplishment of this work in the development of dorsal root ganglia organoids, however, the composition of the organoids was not completely described. Based on the results presented in Figure 2D, about 70% of total cells are neurons (BIII-Tub), which raises the question of what would the remaining 30% be. Single-cell PCR for not only neuronal, but also glial markers should help elucidate this issue.

Answer:

S100 staining analysis predicted that at least 10% of the remaining cells are satellite-like/Schwann-like cells (Figure S5c,d). In accordance, we found the presence of about 10% of p75⁺ cells (Figure S5e,f). Moreover, the presence of *NEUROG1* at DIV40 and DIV80 (Figure S3d) and strong SOX2 expression (Fig. S3e) suggested the presence of immature neural precursors at those stages.

Heatmaps showing the similarities between DRGOs and primary DRG for sensory neurons precursors, Schwann and Satellite cells specific transcript confirmed the presence of these cell types (Figure S5j,k).

To determine the exact cellular composition of DRGOs, we performed 10x Chromium-based single-cell mRNA sequencing to molecularly profile 5,363 single cells (Fig. 3c-j). This analysis allowed us to recognize the entire collection of cells present in the healthy DRGOs showing the presence of the three sensory neuronal subtypes, several progenitor cell groups, satellite and Schwann cells as normally found in primary DRGs.

3. Finally, as described in this study “the removal of exclusively the expanded GAA tract was less effective in reverting the pathological hallmarks with respect to the larger deletion which ablated most of FXN intron-1”. The authors hypothesized that the inability to restore FXN expression in the removal of expanded GAA tract cells was due to epigenetic mechanisms. There are 2 issues that should be addressed in the study of this hypothesis:

a. The authors did not explore the possibility of the shorter deletion impacting the splicing of the intronic region. To exclude this possibility, a qRT-PCR should be performed to show that deletion of repeats alone does not disrupt the splicing of FXN. This should be carried out for all alleles (expanded, control, LD and SD). If negative, this should strengthen the author’s claim regarding the mechanism of FXN repression in the shorter deletion.

Answer:

As suggested by the reviewer, we performed qRT-PCR assays with primers flanking the conjunction between FXN exons 1 and 2. The relative abundance of these

amplicons is directly comparable with the RNA and protein expression levels of the parental and modified patient iPSCs (Figure S7a,b). Accordingly, we did not detect the presence of unspliced products by RT-PCR analysis (data not shown).

These data together with the results obtained with the supplemental epigenomic analysis performed during this revision provide solid evidence that the FRDA-SD iPSC lines and derivatives are unable to restore FXN expression primarily for defected epigenetic mechanisms.

b. It was not clear why the authors chose to investigate H3K9me3 specifically. Therefore, there is a need for an epigenetic screening as H3K9me3 may not be the only repressive factor involved in the FDRA pathogenesis.

Answer:

Initially, we profiled the H3K9me3 chromatin mark since it is among the most studied epigenetic change within the FXN genomic locus. Specifically, di- and tri-methylation of H3K9 are of particular interest since several studies reported increased levels of these epigenetic marks in various FRDA models (Savellev et al. 2003; Herman et al. 2006; Al-Mahdawi et al. 2008). However, we fully agree with the reviewer that other chromatin modifications as well are associated with FXN gene silencing. To extend this analysis, MeDIP-qPCR on the FXN promoter region was performed and confirmed a substantial reduction in DNA methylation in FRDA samples after CRISPR/Cas9 mediated GAA tract deletion. Similarly, the H3K9ac mark was found profoundly reduced in PTL6, but less in PTS36, which was rescued after CRISPR/Cas9-dependent DNA excision (Figure 5g).

Minor Issues:

1. Reference missing from sentence: “The length of the GAA expansion correlates proportionally with the severity of the disease and inversely with the age of onset which can occur from infancy to after 60 years of age.”

Answer:

We added the following reference for this sentence:

Campuzano, V., Montermini, L., Lutz, Y., Cova, L., Hindelang, C., Jiralerspong, S., Trottier, Y., Kish, S.J., Faucheux, B., Trouillas, P., et al. (1997). Frataxin is reduced in Friedreich ataxia patients and is associated with mitochondrial membranes. *Human Molecular Genetics* 6, 1771–1780.

2. The authors described electrophysiological measurements as well as synapsin and vGLUT analysis for characterization of the DRGOs, which raises the question: were there differences between affected and unaffected organoids? Especially, due to the axonal defect reported in figure 6C and 6D.

Answer:

We performed Synapsin immunofluorescence analysis on PTL6 and its isogenic line LD3 at 50 days of coculture with intrafusal fibers. We observed a marked reduction of Synapsin positive puncta along FRDA axons, which is robustly recovered after CRISPR/Cas9 *FXN* intron1 deletion (Figure 7j,k).

3. In the title: “FRDA deficits in Fe-S cluster protein biogenesis and antioxidant response are restored in PTL6-DL DRGOs”, please correct PTL6-DL to PTL6-LD

Answer:

We amended the text accordingly.

4. In the legend of figure 5 E, please introduce the acronyms UC5', UpSD, DwSD, UC3'.

Answer:

We revised the figure legend as following: “Illustration of the genomic sites within *FXN* exon 1 and its downstream intron where H3K9me3 relative abundance was profiled by ChIP-qPCR analysis. EX1, Exon 1; UC5', Uncutted region at 5'; UpSD, Region upstream Short deletion; DwSD, Region downstream Short deletion; UC3', Uncutted region 3'.”

Reviewer #2 (Remarks to the Author):

In their manuscript, Mazzara and colleagues report: 1) the generation of human induced pluripotent stem cells (iPSCs) from two patients with Friedreich's ataxia (FRDA) and GAA repeat expansions of different lengths; 2) the generation of isogenic controls via the CRISPR-mediated excision of the GAA repeats or of most of *FXN* intron 1; the differentiation of these iPSCs into what they call dorsal root ganglia organoids (DRGOs), including the establishment of connections with co-cultured muscle spindles; excess cell death, loss of Fe-S proteins and abnormal antioxidant responses in FRDA-derived DRGOs.

This is an interesting study with potential to provide relevant insight into FRDA pathogenesis. There are however, some concerns:

1. No details are provided about the control iPSC line utilized.

Answer:

Control iPSC clones (CTRL and CTRL2) were derived from skin fibroblasts from a healthy donor with age (45 years) comparable to that of the two FRDA patients. Fibroblasts from both healthy and FRDA individuals were reprogrammed with the CytoTune®-iPS 2.0 Sendai Reprogramming Kit (Invitrogen A16517) and assessed for expression of pluripotency-specific marker, cell colony morphology, in vitro trilineage differentiation potential and karyotype. Additionally, we added the results of the genotyping and repeated-primed PCR assays on the *FXN* GAA expansion, confirming a normal GAA tract in healthy (CTRL1 and CTRL2) and the expanded GAA sequences in the FRDA patient cell clones (Figure S1a,b).

2. The morphological differences between DRGOs and the clusters formed in traditional differentiation protocols for SN should be better specified. In both cases cellular aggregates extending long neurites form.

Answer:

Sensory neurons differentiated from iPSCs following 2D *in vitro* protocols might generate dispersed clusters in the culture with random cell type composition and organization. From this point of view, DRGOs are very different from these clusters since their relative cellular composition is comparable from one another, contain both neuronal and glial cells that over time supporting each other their differentiation. In fact, 2D-differentiated sensory neuronal cultures are mostly lacking glial cells (Chambers et al., 2009; Young et al., 2016), that are well represented in DRGOs. Of note, when we tried to differentiate FRDA patient iPSCs into iPSC-SNs, we failed to obtain double PRPH+/BRN3A+ neurons. In fact, we observed an extensive cell death that made impractical to model FRDA with sensory neurons obtained with classical 2D-differentiation protocols (Figure S3c). A plausible reason for this may be the lack of paracrine signals that mediate the mutual neuronal survival during *in vivo* development which is instead well recapitulated in 3D-organized DRGOs.

3. In FRDA, proprioceptive neurons are most vulnerable, showing developmental defects, while other SN neurons seem to be affected only later (e.g. Creigh P et al. *Ann Clin Transl Neurol* 2019; 6:1718-1727), so it is important to understand the relative representation of different types of SN in the DRGOs. This is given only approximately. In particular, data suggest that a significant number of neurons are TrkB/TrkC double positives, i.e. their fate as proprioceptors or mechanoreceptors is not yet determined. This requires co-localization experiments of specific markers defining such subpopulation. IF images of better quality should be provided at different magnifications. Single-cell RNAseq may also be used. This is important also because most FXN in DRGs comes from proprioceptors (see Piguet F et al. *Mol Ther.* 2018; 26:1940-1952).

Answer:

Prompted by this request of the reviewer, we decided to perform 10x Chromium-based single-cell mRNA sequencing to define the exact cell composition of DRGOs. In the revise manuscript we presented the molecular analysis of 5,363 single cells isolated from healthy DRGOs (Figure 3c-j). UMAP clustering and heatmap gene profiling enabled us to determine the DRGO cellular diversity identifying the three major sensory neuronal classes, several groups of neuronal precursors and the two groups of glial cells, satellite and Schwann cells. With this analysis, we were able to define the relative abundance of the different classes of neurons and glial cells in the DRGOs (Figure 3c-j).

4. The excess cell loss and abnormal morphology of FRDA DRGOs is an interesting observation, but it should be better specified. Is there a differentiation deficit? Does it specifically affect proprioceptors? If normally differentiated cells die prematurely, what are their identities, is there a specific vulnerability?

Answer:

Initially, to answer to this question we decided to perform scRNA-seq from the FRDA DRGOs. Unfortunately, caused by the excessive number of dying cells the quality of the RNA was not sufficient to move forward with the library preparation and following sequencing. This problem did not occur with the analysis of unaffected DRGOs which has been included in the revised manuscript. To anyhow investigate this issue, we decided to perform qPCR analysis for a restricted number of specific molecular markers of the different neuronal cell types on the naïve and treated FRDA DRGOs. Interestingly, this expression profiling suggested that proprioceptive and nociceptive neurons were substantially reduced while mechanoreceptors remained stable during the course of the analysis. Interestingly, the proprioceptor marker PV for was the most reduced in FRDA DRGOs suggesting that this population of neurons might be particularly affected.

5. Electrophysiology data confirm the neuronal phenotype of most analyzed cells, but again it is unclear what type of neurons they represent. Are there any corresponding, morphology data, i.e. after biocytin injection of clamped cells?

Answer:

To answer to this important comment of the reviewer we performed combined patch-clamp recording with qPCR analysis on 50 neurons from 6 DRGOs (Figure 3k-n). Electrophysiology was performed on DRGOs transduced with a virus expressing GFP under the Synapsin promoter for selective live imaging of the sensory neurons (Figure 3k). This analysis confirmed that almost 70% of the GFP+ neurons reached functional maturation with hyperpolarized membrane potential and the ability to spike repetitive action potentials (V_{rest} of $-53,8 \pm 10,3$ mV, $n=50$ neurons) (Figure 3l). Further characterization of the synaptic activity revealed the presence of spontaneous excitatory postsynaptic currents (EPSCs) in approximately 60% of the analyzed neurons (Figure 3M), indicating the formation of a functional network, as suggested by the diffuse presence of synaptic markers and corroborating the functional maturation of our DRGO sensory neurons. Single patched cells were subsequently analyzed by single neuron RT-PCR to identify the specific identity. Expression analysis of subtype specific markers indicate the presence of nociceptive neurons (8%), mechanosensory neurons (44%) and proprioceptive neurons (28%) (Figure 3n). 20% of patched neuronal cells did not expressed any subtype specific marker of the three mature sensory neurons, suggesting that they can represent an immature population of sensory neurons in line with scRNA-sequencing data (Figure 3n).

6. In Figure 4 it is not specified if electrophysiology data are from FRDA, controls or isogenic control cells. These should be analyzed separately.

Answer:

Electrophysiology analysis was performed on DRGO sensory neurons derived from control iPSCs (Figure 3k-n).

7. The LD vs. SD observation is intriguing, but the only possible explanation is the persistence of inhibitory epigenetic marks in intron 1 after removal of GAA expansions. These may be histone PTMs or DNA methylation, which may last longer and has not been explored.

There is some evidence that long repeats may be associated with more extensive DNA methylation in intron 1. The effect of continued culturing of iPSCs allowing more cell divisions should also be investigated.

Answer:

As suggested by the reviewer, we performed analysis of the DNA methylation state by meDIP-qPCR and profiled the H3K9ac chromatin mark.

MeDIP-qPCR confirmed an increased level of DNA methylation on the *FXN* promoter which was substantially reduced in intron-1 deleted FRDA iPSCs (Figure 5h). On the same line, the deposition of H3K9ac appear severely affected in PTL6 FRDA DRGOs and substantially recovered after removal of the expanded GAA tract.

8. There are several imprecisions about the disease, including the statement that proprioceptive neurons and Purkinje cells are most vulnerable (page 6). This is incorrect, as the most vulnerable cells in the cerebellum are large projection neurons in the deep nuclei, while Purkinje cells are mostly preserved in number and morphology, except for their axonal endings in the deep nuclei undergoing grumose degeneration, possibly due to loss of target cells.

Answer:

We thank the reviewer for this comment and we have revised the manuscript quoting these important findings.

9. The paper needs a revision for grammar and style, preferably by a native English speaker.

Answer:

We fully revised the text grammar and style with a support of a native English mother tongue.

Reviewer #3 (Remarks to the Author):

Extremely interesting manuscript in which the authors developed an original and novel approach to generate organoid cultures of DRG sensory neurons from iPSC. This is the first example of a protocol for DRG organoids in the literature. Furthermore, the authors developed a very elegant experimental design to allow maturation of the DRGO in the presence of myogenic culture, and demonstrates the presence of muscle spindle. Overall, these novel experimental approaches could be important to develop human-derived models for sensory neuropathies. In the current manuscript, the approach is applied to iPSC cells derived from Friedreich's Ataxia (FRDA) patients, a disease that primarily affects the proprioceptive neurons of the DRG. There is a current lack for good FRDA neuronal models, and the iPSC derived DRG organoids appears as a very powerful solution as it combines the complex genetic context of FRDA (large GAA repeat expansions) and the primarily affected cell type.

Overall, this work represents an important advance in the field of stem cell research through the elaboration of a new DRG organoid differentiation protocol. There are however several weaknesses that temper the enthusiasm.

Major comments:

Although not completely characterized, it is thought that there are multiple regulatory elements in intron 1 of the frataxin gene. Therefore, removing most of intron 1, as proposed in the manuscript, might have deleterious effect on the fine regulation of frataxin expression.

Answer:

Actually, regulatory elements in *FXN* intron 1 have not been studied in details and few and fragmented information are available up to date. Importantly, the vast majority of identified regulatory sequences controlling *FXN* expression were discovered using plasmid reporters on cell lines. Thus, these results should be taken cautiously since they reflect only marginally the *FXN* expression modulation in vivo. Nonetheless, in vitro studies showed that the 1255 bp sequence upstream to the ATG starting codon recapitulates *FXN* expression in C2C12 muscle cells (Greene et al., 2005). In addition, an Oct-1 binding site about 4.95 kb from the start codon of the *FXN* gene is necessary for normal gene expression and whose deletion results in decreased gene expression in HeLa and BE-M17 cells (Puspasari et al., 2011). Interestingly, regulatory sequences in intron1 are generally located in the vicinity of *FXN* exon1 (Li et al., 2010) and are left untouched even by the larger deletion which starts 1012 bp after the 5' exon/intron junction. Certainly, we cannot exclude that the two intron1 deletions might excise some other putative elements. However, frataxin protein levels in FRDA-LD DRGOs are comparable to those in control DRGO suggesting an extensive rescue without substantial impairment of gene expression in sensory neurons. A more detail analysis will be required to confirm these data also in other cell types, but we feel that this should be done in a future independent study.

In the current manuscript, the authors suggest that full frataxin recovery can only be achieved with the removal of a large portion of the intron 1 (figure 1F-G, text p6). However, other laboratories in several publications have demonstrated that removing

the GAA expansion repeat restores the frataxin levels and prevents the heterochromatin epigenetic profile (Tremblay, Napierala, Gottesfeld laboratories). Furthermore, Lai et al. (2018) generated isogenic iPSC lines that differ only in the length of the GAA repeat and demonstrated the restoration of frataxin expression. Could the authors elaborate on these differences and find a plausible explanation? The characterization of multiple clones for each deletion and each iPSC line would probably clarify this point.

Answer:

In this work we generated two genomic deletions with different sizes enabling us for the first time to determine their specific ability to rescue FXN expression and FRDA deficits on cells with either small or large GAA tract expansions. Our data strongly indicate that the large deletion is the only able to substantially rescue the pathological effects caused by the long GAA expanded tract (PTL: 460/930 GAA repeats). In contrast, both the small and large deletions are sufficient to restore the molecular pathological effects caused by the relative short GAA expansion (PTS: 330/530).

Previously published studies mostly focused on cellular model with a low-medium GAA expanded tract, but overlooked cases with large expansions (i.e. 541/420 repeats in the Tremblay study). Additionally, several studies were carried out utilizing FRDA cells derived from the YG8R mice with a mild GAA expansion of ~190 repeats. The recovery of the phenotype in these cells and in the murine models is likely due to the mild nature of the disease in this case, leading to a less severe phenotype with mild FXN protein reduction. Furthermore, in those cases the frataxin protein increase was not exactly comparable with healthy controls (Tremblay study) or never reach the levels of controls (Napierala study).

In the study by Lai and colleagues (2019), iPSCs were reprogrammed from two FRDA fibroblast cell lines with 330/338 (GM03816) and 420/541 (GM04078) repeats. The GAA expansion was then replaced with a sequence with a physiological number of GAA repeats by HdAV-assisted homologous recombination in iPSCs. This system can only work in proliferating cells and, thus, cannot be exploited directly into post-mitotic cells, such as neurons. Conversely, our approach mediates the excision of *FXN* intron 1 and is applicable to all type of cells with high efficiency both in *in vitro* and *in vivo*.

The authors state that “PTL6 iPSCs showed the strongest frataxin reduction both at transcript and protein levels compared to PTS36”. However, the western blots and their quantification on Figure 1F-G are in disagreement with this statement. Indeed, frataxin levels appear to be down by approximately 60% in both lines, and if there is a difference, PTS36 appears to have less than PTL6. At the RNA level, there is indeed a clear difference. The authors should address this discrepancy. Again, analyzing several clones might help in clarifying. At the RNA level, PTS36 appears to have 50% frataxin, levels equivalent to heterozygote carriers. At the protein level, both patients have approximately 40% frataxin, inconsistent with the 5-20% reported in the literature. Can the authors elaborate on this aspect?

Answer:

This analysis is referring to *FXN* RNA and protein levels specifically in undifferentiated and proliferating iPSCs. These are not primarily cells affected by the disease and not the focus of this study. Our findings indicate that in this cell type there is no straight and uniform correlation between RNA and protein levels. Further analysis would be necessary to investigate the basis for this behavior which, however, remains beyond and out of the focus of the present study.

The authors convincingly generated DRG organoids that are composed of all neuronal cell types. A technical question: what is the effect of the anti-mitotic treatment on glial cells, considering that these are extremely important for signaling and for myelination?

Answer:

The anti-mitotic treatment was intended to reduce the number of proliferating precursors and force differentiation in post-mitotic neurons. Nonetheless, this also causes a reduction in peripheral glial precursors, including Schwann cell progenitors. The anti-mitotic agent is particularly useful when organoids are generated for electrophysiological studies. In fact, since satellite cells form a monolayer sheath around the neuronal soma, the removal of these cells facilitates membrane access for patch clamp analysis. It should be noted, however, that this brief treatment does not rid the DRGO entirely of S100+ glial cells but simply reduced the number of mitotic cells. Indeed, we were able to observe an average of 10% of S100+ glial cells in the DRGOs, which is still sufficient for sustaining neuronal survival and maturation.

The immunofluorescence images from 2B and 2C show very convincingly that the cultures are well organized and contains large sensory neurons that are positive for NF200. Considering that the RT-PCR is not quantitative, it is difficult to evaluate to which extent the culture is enriched for any of the 3 neuronal subtypes. The author should perform quantitative RT-PCR or Western Blot experiments to evaluate the enrichment. This statement holds for all the RT-PCR presented in the manuscript, quantitative would be better (Fig. 2, S2, S3). Furthermore, a better choice of specific antibodies or RT-PCR to determine the population of sensory neurons present in their culture would strengthen the manuscript. Indeed, the authors chose Runx3 and TrkC (NTRK3). Runx3 is specific of proprioceptive neurons during development and its expression stops when the neurons are mature (Marmigère and Ernfors, 2007). TrkC is expressed both by mechanoreceptive and proprioceptive neurons, as reported in Usoskin et al., 2015. Usoskin et al. present new proprioceptive specific genes such as CNTNAP2 and SPP1 that should be used.

Answer:

In line with this request and in order to obtain a precise description of the cellular composition of the DRGOs we embarked in scRNA sequencing profiling 5,363 cells isolated from 3 DRGOs (Figure 3c-j). Moreover, we performed single cell qPCRs for selective and predictive markers of cell type identity on 50 neurons which were

previously patched-clamp recorded from 6 different DRGOs (Figure 3k-n). Both studies obtained similar results and allowed us to obtain a precise analysis of the different neuronal and glial cell types composing the DRGOs, confirming the presence of the three main sensory neural subtypes as well as satellite and Schwann cells. Importantly, unsupervised UMAP clustering correctly identified the three main subtypes of sensory neurons and different classes of neural progenitors together with satellite and Schwann cells (Figure 3c-n).

While there is no doubt that the DRGO contain all subtypes of DRG sensory neurons, the authors should select images with a better staining pattern for the different markers (Figure 2G). Based on the images, it is difficult to understand how the authors manage to quantify properly the subtypes, considering the large amount of overlap in the staining. Furthermore, it is difficult to understand how the total percentage can be above 100% (Figure 2H – 114%), except if several neurons express two different receptors (as expected for mechanoreceptive neurons). The percentages given in the text (p. 7-8) do not correspond to the percentage shown on the graph Figure 2H. Indeed, the text states that the highest percentage is 44.5% corresponding to NTRK2, while Figure 2H shows that NTRK3 has the highest percentage. Is there an inversion? 44.5% of neurons are positive for TrkC which is possible since it stains both proprioceptive and mechanoreceptive neurons. It is however incorrect to state that 44.5% are proprioceptive. The percentage of proprioceptive would be the TrkC positive part (44.5%) minus the TrkB positive part (exclusively mechanoreceptive) (35.8%), which would correspond to around 8-9%. Parvalbumin is specific for proprioceptive neurons in the DRG, and Figure 3E-F shows around 15% of Parvalbumin positive neurons, a percentage that is closer to the physiological number. Overall, the transcriptional profile demonstrates nicely that the DRGO present an expression profile similar to DRG albeit probably closest to developmental stages and immature.

Answer:

In line with the reviewer comments, we extensively revised this part of the work, by implementing a wide scRNA-seq analysis of 5,363 cells to identify and classify the different cell types in DRGOs. This profiling allowed us to distinguish proprioceptors, nociceptors and mechanoceptors for the expression of specific molecular markers and by unsupervised UMAP clustering. The new data are presented in Figure 3c-j and discussed in the text.

The results obtained in figure 4 are clearly a highlight of the paper, showing the capacity to generate mature DRG organoids capable of muscle spindles forming and that are electrically active. These extremely impressive and innovative results could be strengthened by complementary experiments. Indeed, it would be interesting to identify the type of neurons that forms the muscle spindle, since in the physiological context, both motor and sensory neurons are found. This could be done using a Parvalbumin antibody to demonstrate the presence of proprioceptive neurons.

Answer:

Because the organoid is a highly compacted cell aggregate it would be extremely challenging to trace an axon back to its soma within the packed cellular core of the organoid. For this reason, the neuronal identification by PV co-staining is not doable in these conditions. However, we can confidently identify these neurons morphologically. It is well known that sensory and motor neurons contact the intrafusal muscle fibers in different ways: motor neurons contact the extinctions of the intrafusal fibers, while proprioceptive neurons contact the central region where the nuclei are allocated. Interestingly, we observed that axonal terminals form annulospiral wrapping around the nuclei region of the intrafusal fibers, as it is normally accomplished by the sensory component of the muscle spindle. It has to be also mentioned that DRGOs do not contain motor neurons as confirmed by the scRNA-seq analysis and immunostaining for ISL1.

With the current protocol, are muscle spindles sparse or are they found in a reasonable number?

Answer:

With the protocol described in our study, we observed that 30 to 40% of the total plated muscle cells developed multinucleated fibers (data not shown). Of these, 50% represent the population of intrafusal fibers (Figure S5b,c). After the coculture, among these intrafusal fibers, those that contact the sensory axons survive long term and of these it was possible to analyze only those most in the periphery to avoid that too many axons overlap. With this premise, the number of analyzable muscle spindles ranges from 0 to 9 each random field at 10x magnification.

Although probably beyond the current manuscript, have the authors tried to myelinate the neurons to achieve better maturation and functionality?

Answer:

This is a very important question and we are very interested to follow this up. However, in our first attempts we found that the time necessary for myelination is extremely long (months) and the culture system needs to be adapted to maintain an optimal number of cells in the cultures with a correct ratio between the different cell types over this very protracted time course. Then, we will need exceedingly more time to properly achieve some significant data on this particular aspect.

Figures 5A to 5D show conflicting results with the mRNA and protein expression quantifications inversely correlated. The long repeat line (PTL6) shows lower protein level than the short repeat line (PTS36), as expected (although opposite to the result on the IPSC lines). Unexpectedly, the mRNA quantifications show the opposite. The authors should explain these conflicting results.

Answer:

We apologize with the reviewer for this obvious mistake occurred inadvertently during the figure assembly. In fact, we have mistakenly swap the PTL6 and its isogenic lines data with the PTS36 and its isogenic lines results. The images have been correctly reshuffled in the revised Figure 5.

The protein levels in the PTL6 line are this time in agreement with the levels described in the literature for FRDA patients. The effect of the SD or LD excision on the expression levels of frataxin appears not to be the same as for the iPSC lines, in particular for PTS36-SD23. Furthermore, high variability is seen – could this be a reflection of variation in the differentiation protocol with variable percentage of neurons from one organoid to another? The authors should discuss this issue in the manuscript.

Answer:

We agree with the reviewer on this comment. This is likely due to the fact that DRGOs contains a sizeable amount of immature neurons in the process of maturation and this cellular component can vary among organoid. This cell variability might be the cause for difference in *FXN* gene expression.

In Figure 5F-G, the H3K9me3 pattern of PTL6 is different between the two figures. For example, in 5G, at the UC5' site, the level of the mark is higher than any other site, although it is among the lowest in 5F. Could the authors comment these results?

Answer:

The first histogram is referred to “% of enrichment on input” instead of “Relative to H3”, while the second histogram is “relative to H3”. We have standardized the results to “% of enrichment on input” and added the analysis for H3K9ac and meDIP-seq to confirm our results.

It appears that the authors accidently present the exact same graphs in 6B as this is not in agreement with the rest (i.e. should the first graph be PTS36 and derivatives?). It would be worth for the authors to check.

Answer:

We apologize for this inconvenience. We have corrected this in the revised Figure 6.

Overall Figure 6 shows a nice rescue of the phenotypes after the GAA repeats excision. However, there is a lack in tissue specificity, as the result show that every cell type is affected by *FXN* depletion, although proprioceptive neurons should be more sensitive than the rest (as explained earlier by the authors). Could the authors comment on this? Clearly the phenotype observed in the dish appears to agree with a developmental component of the disease as proposed more recently in the field. However, the authors also see an increase in death in FRDA DRGO compared to controls. In the text p 12, it is not always clear to which control the authors are referring to. This section should be re-written.

Answer:

The control refers to healthy donor iPSC-derived DRGOs. The text was amended as following: "We observed a marked increase in disorganized and apparently dying PTL6 DRGO cells with respect to control cell lines ($61,5\% \pm 5,3\%$ and $95,8\% \pm 3,6\%$, respectively). In contrast, the fraction of seemingly healthy DRGO cells was restored to control levels in both PTL6-SD10 and -LD3 DRGOs ($96,9\% \pm 7,9\%$ and $95,9\% \pm 3,3\%$, respectively). Similar results were found at DIV 40, where the elevated number of dying DRGO cells of patient cell lines compared to control DRGOs ($27,4\% \pm 5,8\%$ and $8,2\% \pm 3,4\%$, respectively) was extensively rescued in both PTL6-SD10 and PTL6-LD3 DRGOs ($10 \pm 3,7\%$ and $9\% \pm 4,2\%$, respectively), as highlighted by cleaved Caspase 3 staining (Figure 6a,b).

The authors explore the phenotype of the FRDA DRGO, with a particular focus on mitochondria, as it is known that mitochondria are particularly susceptible to FXN depletion. The authors have well adapted a powerful technique long known to arrange neuronal axons in microgrooves, allowing to follow organelles movement and follow the direction of the movement, using microfluidic chips. The novelty here is that the authors have designed additional chambers compared to classic neuron adapted microfluidic chips, that permits the cultures of the DRGO. With this system, the authors were able to measure several parameters of mitochondrial morphology and dynamics. The authors demonstrated a decreased mitochondrial number (confirmed by mtDNA/gDNA quantifications in supplementary data), and overall decreased mitochondrial size. The graph and statistics for the circularity should be checked because there does not appear to be any difference between the 3 conditions.

Answer:

As much as it complies with the bar graph in Figure 7f, the number of counted mitochondria is extremely high and, thus, sufficient to indicate that also a small change in circularity is indeed significant as we outlined in the bar graph. However, we expanded even further the counting with new samples which confirmed and further strengthened the different between the genotypes.

In the last part of the manuscript, the authors investigated FRDA-related phenotypes such as iron-sulfur clusters (ISC) deficiency and oxidative stress. This is by far the weaker part of the manuscript and would benefit from being reinforced by a few complementary experiments. It is important to demonstrate whether there is an ISC cluster deficiency has this is the primary role of frataxin. The authors demonstrate a very strong decreased level of aconitase (around 90% in the PTL6 mutant), which seems unlikely as this protein is known to be the less sensitive to degradation when deprived of its ISC. There are numerous other proteins that can be tested by western blot to demonstrate indirectly a ISC deficiency, such as Complex I and II of the respiratory chain or Lipoic Acid bound to proteins. The activity of the enzyme (such as SDH or aconitase which are relatively standard measurements) would constitutes a

more direct way to evaluate ISC loss. Increase in SOD1 and SOD2 mRNA are interesting results, but are not enough to conclude that there is the presence of an increase oxidative stress in the model. Staining with fluorescent probes specific to oxidative stress such as DHR123 or MitoSOX should be perform. Furthermore, using such probe could allow to investigate the phenotype specifically in the different subtypes of neurones. The tissue specificity is an important question in the FRDA field. Furthermore, the authors could also use a fluorescent probe that allows to investigate mitochondrial health status and activity such as TMRM, TMRE or JC-1. Additional information regarding iron metabolism would be much appreciated as it is part of the classical hallmarks of FRDA.

Answer:

We evaluate the cellular oxidative stress focusing on Reactive Oxygen Species (ROS) through the sensitive fluorescent probe 5-(and-6)-chloromethyl-2',7'-dichlorodihydrofluorescein diacetate (CM-DCF). We confirmed a reduction in ROS content in the FRDA-LD compared to unmodified FRDA DRGOs, as shown in Figure S5e and discussed in the manuscript.

Minor comments:

1. The bibliography in the introduction is very sparse and an effort should be made to cite primary bibliography.

Answer:

We changed the text according to this remark of the reviewer.

2. The introduction is not completely updated with the latest development in the FRDA field that are important in the context of the current manuscript

Answer:

We modified the text according to this remark of the reviewer.

3. Cerebellar Purkinje cell are not extremely affected in FRDA, it is rather the dentate nucleus of the cerebellum that is affected. This should be corrected (p. 6)

Answer:

We apologize with the reviewer for this inaccuracy that has been corrected in the revised manuscript.

4. A throughout reading of the manuscript should be done to remove the multitude of typos or small annotation mistakes that can be found (some examples: legend figure 1B, "FRDA-DS and FRDA-DL" instead of FRDA-SD and FRDA-LD; legend figure 1, "organoids" instead of IPSC; Figure S1E, the oligos used are probably LF1+LR+SR and not LF2 as noted;)

Answer:

We amended the typos that the reviewer identified throughout the text. The primed utilized are LF1 + LF2 + LR and not SR as correctly pointed out. Moreover, organoid instead of iPSCs has been added in legend of Figure 1.

5. Figure 3B – it is extremely difficult to discriminate between the different blues and green, and therefore it is difficult to discriminate between DRGO and IPSC. The colors should be changes

Answer:

Following the reviewer's suggestion, we changed the PCA colors

6. The manuscript presents data on 6 different IPCS lines. For the graphs, it would be important to have a code which could differentiate between the different lines. This would considerably help. For example, PTL6 is grey in 5F and then red in 5G. Maybe using hatched bars for one of the lines would help, and color code could then stay.

Answer:

We thank the reviewer for this comment and modified accordingly the graphics with an easy-to-follow color code for either of the two patient cell lines and their isogenic treated derivatives.

REVIEWERS' COMMENTS:

Reviewer #1 (Remarks to the Author):

The majority of the concerns have been addressed satisfactory and the overall quality over the work has improved.

Reviewer #2 (Remarks to the Author):

The issues raised about the previous version of this paper have largely been addressed in this revision. Two minor points:

1. The paragraph in the introduction about mechanisms of epigenetic suppression of FXN expression needs some improvement. There is no clear link between non-B structures formed by long GAA repeats in vitro and epigenetic mechanisms, so their role in vivo remains hypothetical. Epigenetic mechanisms include repressive (rather than abnormal) histone marks such as loss of acetylation and increased di-trimethylation at both H3K9 and H3K27, formation of D-loops, and DNA methylation. As these changes are explored in the cell model described in the paper, they should more clearly summarized in the introduction. Also, animal models (e.g. YG8 mice) in addition to cell models and patient samples have shown these changes.
2. The statements that that "2D iPSC-SNs form unorganized clusters in a random pattern with unproven participation of different neuron subtypes, fail to recapitulate the DRGs spatial architecture and cellular diversity" and "when we tried to differentiate FRDA patient iPSCs into iPSC-SNs, an extensive cell death occurred such that we were unable to obtain neuronal cultures of decent quality with enough double PRPH+/BRN3A+ sensory neurons on which to perform studies" should be tempered in the light of several reports of SN differentiation from iPSCs, including from FRDA patients, which turned out to be viable and of controlled composition (e.g. Lai et al. JBC 2019; 294:1846-1859 and most recently Dionisi et al. Sci Rep 2020; 10:7752). This does not affect the importance, interest, power and elegance of the 3D model developed by the authors, even if 2D models are also effective.

Reviewer #3 (Remarks to the Author):

The revised version of the manuscript is much improved and the authors have answered to most of the major and minor comments. The addition of the scRNA sequencing of the control DRGO brings a wealth of information.

There are still a few minor comments that should be addressed:

1. The molecular characterization of the FRDA DRGO is very preliminary, and would benefit from further investigation before making statements such "Thus, these results demonstrate that diseased DRGOs recapitulated many peculiar aspects of the pathological phenotype present in patient somatic cells. Furthermore, the response to oxidative stress in both the mitochondrial and cytoplasmic compartments and the impairment in the generation of Fe-S proteins...". From the data presented, there is no evidence of impairment of Fe-S cluster biogenesis. The deficiency of aconitase appears to be at the transcriptional level and is most likely a reflection of the decrease biogenesis of mitochondria. It is understandable that the authors might consider the deep characterization of the molecular phenotype as a story beyond the current manuscript (and this is comprehensible). However, the strong affirmative statements in the results and to a lesser extend in the discussion should be revised to reflect the results obtained.
2. There are a few mistakes in the figures (e.g. legend of Fig.S1 does not correspond to what is seen in Figure p33 line 1078, no * on Fig. 3b)
3. There are still a few typos throughout the manuscript (e.g. p11, line 354, p 30 line 974)

July 27^h, 2020

Manuscript #NCOMMS-19-29226R1 entitled “CRISPR/Cas9 targeted deletions rescue Friedreich’s ataxia pathological deficits in a stem cell-based sensory neural circuitry with dorsal root ganglia organoids“ by Mazzara, Muggeo et al.

Reviewers' comments:

Reviewer #1 (Remarks to the Author):

The majority of the concerns have been addressed satisfactory and the overall quality over the work has improved.

Response:

We thank the reviewer for this positive comment on our work.

Reviewer #2 (Remarks to the Author):

The issues raised about the previous version of this paper have largely been addressed in this revision.

Two minor points:

1. The paragraph in the introduction about mechanisms of epigenetic suppression of FXN expression needs some improvement. There is no clear link between non-B structures formed by long GAA repeats in vitro and epigenetic mechanisms, so their role in vivo remains hypothetical. Epigenetic mechanisms include repressive (rather than abnormal) histone marks such as loss of acetylation and increased di-trimethylation at both H3K9 and H3K27, formation of D-loops, and DNA methylation. As these changes are explored in the cell model described in the paper, they should more clearly summarized in the introduction. Also, animal models (e.g. YG8 mice) in addition to cell models and patient samples have shown these changes.

Response:

In line with this observation of the reviewer, we have modified in the Introduction the description of the mechanisms leading to FXN silencing as follows:

“The FXN gene silencing is speculated to result from the formation of non-B DNA structures such, as triplexes, or persistent DNA-RNA hybrid structures which might impede transcription initiation and elongation⁸. However, studies performed in FRDA animal and cellular models, then confirmed in human heart, cerebellum and brain explants, indicated repressive histone modifications in the intronic regions flanking the GAA expansion as an additional cause of gene silencing⁹⁻¹¹. Particularly, increased di- and trimethylation of H3K9 and deacetylation of histones H3 and H4 at lysine residues are present around the expanded repeat tract suggesting that heterochromatin-mediated transcriptional repression is one the main cause of FXN silencing”.

2. The statements that that “2D iPSC-SNs form unorganized clusters in a random pattern with unproven participation of different neuron subtypes, fail to recapitulate the DRGs spatial architecture and cellular diversity” and “when we tried to differentiate FRDA patient iPSCs into iPSC-SNs, an extensive cell death occurred such that we were unable to obtain neuronal cultures of decent quality with enough double

PRPH+/BRN3A+ sensory neurons on which to perform studies” should be tempered in the light of several reports of SN differentiation from iPSCs, including from FRDA patients, which turned out to be viable and of controlled composition (e.g. Lai et al. JBC 2019; 294:1846-1859 and most recently Dionisi et al. Sci Rep 2020; 10:7752). This does not affect the importance, interest, power and elegance of the 3D model developed by the authors, even if 2D models are also effective.

Response:

We changed the text as follows:

“2D-differentiation strategies have been employed to generate sensory neurons from iPSCs, but these do not form DRG-like structures. It is reported that 2D iPSC-SNs form unorganized clusters in a random pattern with unproven concomitant participation of different neuronal subtypes and glial cells, fail to recapitulate the DRGs spatial architecture and cellular diversity^{35,36}. Of note, even if some recent studies showed new derivation of FRDA SNs by direct cell reprogramming or iPSC differentiation^{37,38}, when we tried to generate SNs from FRDA patient iPSCs, an extensive cell death occurred which prevented us to obtain neuronal cultures with sufficient numbers of PRPH+/BRN3A+ sensory neurons (Supplementary Fig. 3c). We hypothesized that the absence of glial cells and sufficient cell-to-cell contacts in 2D cultures may further reduce viability of FRDA iPSC-SN”.

Reviewer #3 (Remarks to the Author):

The revised version of the manuscript is much improved and the authors have answered to most of the major and minor comments. The addition of the scRNA sequencing of the control DRGO brings a wealth of information.

Response:

We thank the reviewer to appreciate the new data included in the revised manuscript.

There are still a few minor comments that should be addressed:

1. The molecular characterization of the FRDA DRGO is very preliminary, and would benefit from further investigation before making statements such “Thus, these results demonstrate that diseased DRGOs recapitulated many peculiar aspects of the pathological phenotype present in patient somatic cells. Furthermore, the response to oxidative stress in both the mitochondrial and cytoplasmic compartments and the impairment in the generation of Fe-S proteins...”. From the data presented, there is no evidence of impairment of Fe-S cluster biogenesis. The deficiency of aconitase appears to be at the transcriptional level and is most likely a reflection of the decrease biogenesis of mitochondria. It is understandable that the authors might consider the deep characterization of the molecular phenotype as a story beyond the current manuscript (and this is comprehensible). However, the strong affirmative statements in the results and to a lesser extend in the discussion should be revised to reflect the results obtained.

Response:

Our conclusions are based on multiple observations both at transcriptomic level for the pivotal genes of the oxidative stress response SOD1, SOD2 and the total reduction in oxidative stress as shown by testing direct live imaging quantification (DCF analysis, Fig. s9f). In addition, our analysis non only assessed

mRNA, but also protein levels for ACO2 (Fig. s9e) and in both cases isogenic treated lines showed a rescue respect to the reduced levels present in patient parental DRGOs.

Thus, we adapted the manuscript as following:

“Thus, these results demonstrate that diseased DRGOs recapitulated many peculiar aspects of the pathological phenotype present in patient somatic cells. Furthermore, the response to oxidative stress in both the mitochondrial and cytoplasmic compartments, represented by the overexpression of SOD1 and SOD2, and the impairment in the generation of Fe-S protein biogenesis, as shown for ACO2, in patient parental DRGOs, were strongly rescued in isogenic PTL6-LD3 DRGOs. Finally, PTL6-LD3 DRGOs presented a reduction of ROS accumulation compared to their parental line.”

2. There are a few mistakes in the figures (e.g. legend of Fig.S1 does not correspond to what is seen in Figure p33 line 1078, no * on Fig. 3b)

Response:

We corrected this inaccuracy.

3. There are still a few typos throughout the manuscript (e.g. p11, line 354, p 30 line 974)

Response:

We amended these inaccuracies.

Kindest regards,

Vania Broccoli
Research Unit Director
“Stem Cells and Neurogenesis” Unit
Division of Neuroscience
San Raffaele Scientific Institute
Via Olgettina 58, 20132 Milan, Italy